# DENSE GAUSSIAN PROCESSES FOR FEW-SHOT SEGMENTATION

## ABSTRACT

Few-shot segmentation is a challenging dense prediction task, which entails segmenting a novel query image given only a small annotated support set. The key problem is thus to design a method that aggregates detailed information from the support set, while being robust to large variations in appearance and context. To this end, we propose a few-shot segmentation method based on dense Gaussian process (GP) regression. Given the support set, our dense GP learns the mapping from local deep image features to mask values, capable of capturing complex appearance distributions. Furthermore, it provides a principled means of capturing uncertainty, which serves as another powerful cue for the final segmentation, obtained by a CNN decoder. Instead of a one-dimensional mask output, we further exploit the end-to-end learning capabilities of our approach to learn a high-dimensional output space for the GP. Our approach sets a new state-of-the-art for both 1-shot and 5-shot FSS on the PASCAL-$5^i$ and COCO-$20^i$ benchmarks, achieving an absolute gain of $+14.9$ mIoU in the COCO-$20^i$ 5-shot setting. Furthermore, the segmentation quality of our approach scales gracefully when increasing the support set size, while achieving robust cross-dataset transfer.

## 1 INTRODUCTION

Image few-shot segmentation (FSS) of semantic classes (Shaban et al., 2017) has received increased attention in recent years. The aim is to segment novel *query* images based on only a handful annotated training samples, usually referred to as the *support set*. The FSS method thus needs to extract information from the support set in order to accurately segment a given query image. The problem is highly challenging, since the query image may present radically different views, contexts, scenes, and objects than what is represented in the support set.

The core component in any FSS framework is the mechanism that extracts information from the support set to guide the segmentation of the query image. However, the design of this module presents several challenges. First, it needs to aggregate detailed yet generalizable information from the support set, which requires a flexible representation. Second, the FSS method should effectively leverage larger support sets, achieving scalable segmentation performance when increasing its size. While perhaps trivial at first glance, this has proved to be a major obstacle for many state-of-the-art methods, as visualized in Fig. 1. Third, the method is bound to be queried with appearances not included in the support set. To achieve robust predictions even in such common cases, the method needs to assess the relevance of the information in the support images in order to gracefully revert to e.g. learned segmentation priors when necessary.

We address the aforementioned challenges by densely aggregating information in the support set using Gaussian Processes (GPs). Specifically, we use a GP to learn a mapping between dense local deep feature vectors and their corresponding mask values. The mask values are assumed to have a jointly Gaussian distribution with covariance based on the similarity between the corresponding feature vectors. This permits us to extract detailed relations from the support set, with the capability of modeling complex, non-linear mappings. As a non-parametric model, the GP further effectively benefits from additional support samples, since all given data is retained. As shown in Fig. 1, the segmentation accuracy of our approach improves consistently with the number of support samples. Lastly, the predictive covariance from the GP provides a principled measure of the uncertainty based on the similarity with local features in the support set.

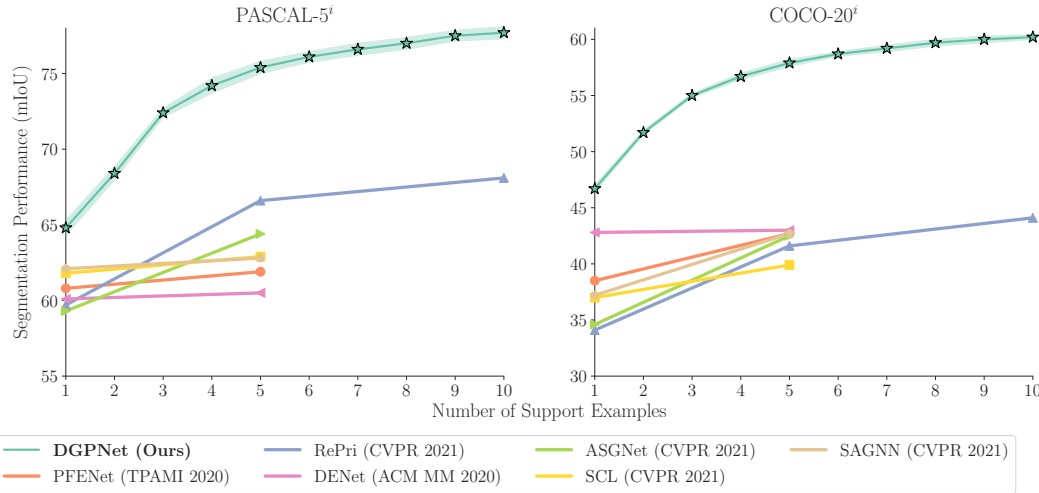

Figure 1: Performance of the proposed DGPNet approach on the PASCAL-$5^i$ and COCO-$20^i$ benchmarks, compared to the state-of-the-art. We plot the mIoU (higher is better) for different number of support samples. For our approach, we show the mean and standard deviation over 5 experiments. Our GP-based method effectively leverages larger support sets, achieving substantial improvements in segmentation accuracy. Our method also excels in the extreme one-shot case, even outperforming all previously reported results on COCO-$20^i$ for any support size.

Our FSS approach is learned end-to-end through episodic training, treating the GP as a layer in a neural network. This further enables us to learn the output space of the GP. To this end, we encode the given support masks with a neural network in order to achieve a multi-dimensional output representation. In order to generate the final masks, our decoder module employs the predicted mean query encodings, together with the covariance information. Our decoder is thus capable of reasoning about the uncertainty when fusing the predicted mask encodings with learned segmentation priors. Lastly, we further improve our FSS method by integrating dense GPs at multiple scales.

We perform comprehensive experiments on two benchmarks: PASCAL-$5^i$ (Shaban et al., 2017) and COCO-$20^i$ (Nguyen & Todorovic, 2019). Our proposed DGPNet outperforms existing methods for 1-shot and 5-shot by a large margin, setting a new state-of-the-art on both benchmarks. When using the ResNet101 backbone, our DGPNet achieves an absolute gain of $14.9$ for 5-shot segmentation on the challenging COCO-$20^i$ benchmark, compared to the best reported results in the literature. We further demonstrate the cross-dataset transfer capabilities of our DGPNet approach from COCO-$20^i$ to PASCAL and perform detailed ablative studies to probe the effectiveness of our contributions.

## 2 RELATED WORK

**Few-Shot Segmentation** The earliest work in few-shot segmentation (FSS), by Shaban et al. (2017), proposed a method for predicting the weights of a linear classifier based on the support set, which was further built upon in later works (Siam et al., 2019; Liu et al., 2020a; Boudiaf et al., 2021). Instead of learning the classifier directly, Rakelly et al. (2018) proposed to construct a global conditioning prototype from the support set and concatenate it to the query representation, with several subsequent works (Dong & Xing, 2019; Zhang et al., 2020; Wang et al., 2019; Nguyen & Todorovic, 2019; Zhang et al., 2019b; Liu et al., 2020c; Azad et al., 2021; Liu et al., 2020b; Xie et al., 2021; Wang et al., 2021). A major limitation of these methods is the *unimodality* assumption. To alleviate this problem, Zhang et al. (2021) construct additional prototypes by a self-guided module, while Yang et al. (2020a); Liu et al. (2020c); Li et al. (2021) instead cluster multiple prototypes to create a richer representation. However, clustering introduces extra hyperparameters, such as the number of clusters, as well as optimization difficulties. In contrast, our method is not restricted in any such sense, and only requires us to choose an appropriate kernel. Some recent works consider pointwise correspondences between the support and query set. These works have mostly focused on attention or attention-like mechanisms (Zhang et al., 2019a; Yang et al., 2020b; Hu et al., 2019; Tian

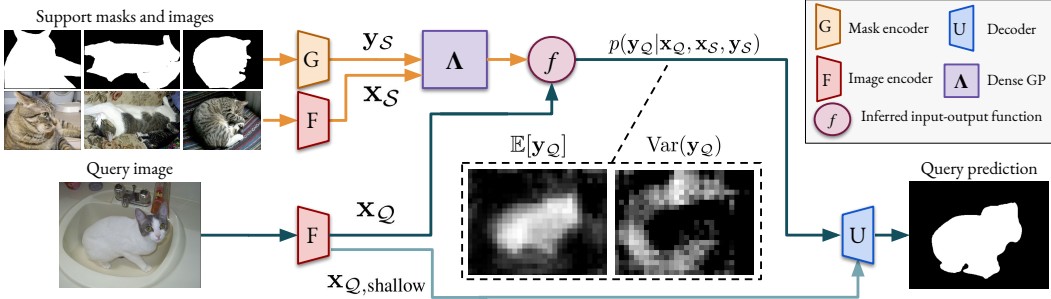

Figure 2: Overview of our approach. Support and query images are fed through an encoder to produce deep features $\mathbf{x}_{\mathcal{S}}$ and $\mathbf{x}_{\mathcal{Q}}$ respectively. The support masks are fed through another encoder to produce $\mathbf{y}_{\mathcal{S}}$. Using Gaussian process regression, we infer the probability distribution of the query mask encodings $\mathbf{y}_{\mathcal{Q}}$ given the support set and the query features (see equations 5-7). We create a representation of this distribution and feed it through a decoder. The decoder then predicts a segmentation at the original resolution.

et al., 2020; Wang et al., 2020). In contrast with these methods, we construct a principled posterior over functions, which greatly aids the decoder.

**Combining GPs and Neural Networks**  While early work focused on combining GPs and neural networks in the standard supervised classification setting (Salakhutdinov & Hinton, 2009; Wilson et al., 2016; Calandra et al., 2016), there has recently been an increased interest in utilizing Gaussian processes in the context of *few-shot classification* (Patacchiola et al., 2020; Snell & Zemel, 2021). Previous works employ the GP in the classification setting and as the final *output layer* in the network, and optimize proxies of either the predictive or marginal log likelihood directly. Note that the classification likelihood is non-gaussian, and hence computing the exact posterior and marginal likelihoods of the model becomes intractable. Here, we go beyond this limitation and propose an internal *dense* GP model of the support features, where the posterior predictive distribution is used as input to a CNN decoder. Moreover, this allows us to learn the output space of the GP to further increase its expressive power. To the best of our knowledge, we are the first to introduce a dense GP approach for the challenging dense prediction task of few-shot segmentation.

## 3 METHOD

### 3.1 FEW-SHOT SEGMENTATION

Few-shot segmentation is a *dense* few-shot learning task (Shaban et al., 2017). The aim is to learn to segment objects from novel classes, given only a small set of annotated images. A single instance of this problem, referred to as an *episode*, comprises a small set of annotated samples, called the *support set*, and a set of samples on which prediction is to be made, the *query set*. Formally, we denote the support set as $\{(I_{\mathcal{S}k}, M_{\mathcal{S}k})\}_{k=1}^{K}$, comprising $K$ image-mask pairs $I_{\mathcal{S}k} \in \mathbb{R}^{H_0 \times W_0 \times 3}$ and $M_{\mathcal{S}k} \in \{0,1\}^{H_0 \times W_0}$. A query image is denoted as $I_{\mathcal{Q}} \in \mathbb{R}^{H_0 \times W_0 \times 3}$ and the aim is to predict its corresponding segmentation mask $M_{\mathcal{Q}} \in \{0,1\}^{H_0 \times W_0}$.

To develop our approach, we first provide a general formulation for addressing the FSS problem, which applies to several recent methods, including prototype-based (Li et al., 2021; Liu et al., 2020a) and correlation-based (Tian et al., 2020; Wang et al., 2020) ones. Our formulation proceeds in three steps: feature extraction, few-shot learning, and prediction. In the first step, deep features are extracted from the given images,

$$\mathbf{x} = F(I) \in \mathbb{R}^{H \times W \times D} \quad . \tag{1}$$

These features provide a more disentangled and invariant representation, which greatly aids the problem of learning from a limited number of samples.

The main challenge in FSS, namely how to most effectively leverage the annotated support samples, is encapsulated in the second step. As a general approach, we consider a learner module $\Lambda$ that

employs the support set in order to find a function $f$, which associates each query feature $x_{\mathcal{Q}} \in \mathbb{R}^D$ with an *output* $y_{\mathcal{Q}} \in \mathbb{R}^E$. The goal is to achieve an output $y_{\mathcal{Q}}$ that is strongly correlated with the ground-truth query mask, allowing it to act as a strong cue in the final prediction. Formally, we express this general formulation as,

$$f = \Lambda(\{\mathbf{x}_{\mathcal{S}k}, \mathbf{M}_{\mathcal{S}k}\}_k), \quad y_{\mathcal{Q}} = f(x_{\mathcal{Q}}) \ . \tag{2}$$

The learner $\Lambda$ aggregates information in the support set $\{\mathbf{x}_{\mathcal{S}k}, \mathbf{M}_{\mathcal{S}k}\}_k$ in order to predict the function $f$. This function is then applied to query features equation 2. In the final step of our formulation, output from the function $f$ on the query set is finally decoded by a separate network as $\widehat{M}_{\mathcal{Q}} = U(\mathbf{y}_{\mathcal{Q}}, \mathbf{x}_{\mathcal{Q}})$ to predict the segmentation $\widehat{M}_{\mathcal{Q}}$.

The general formulation in equation 2 encapsulates several recent approaches for few-shot segmentation. In particular, prototype-based methods, for instance PANet (Wang et al., 2019), are retrieved by letting $\Lambda$ represent a mask-pooling operation. The function $f$ thus computes the cosine-similarity between the pooled feature vector and the input query features. In general, the design of the learner $\Lambda$ represents the central problem in few-shot segmentation, since it is the module that extracts information from the support set. Next, we distinguish three key desirable properties of this module.

## 3.2 MOTIVATION

As discussed above, the core component in few-shot segmentation is the few-shot learner $\Lambda$. Much research effort has therefore been diverted into its design (Nguyen & Todorovic, 2019; Liu et al., 2020a; Wang et al., 2020; Liu et al., 2020c; Yang et al., 2020a; Li et al., 2021). To motivate our approach, we first identify three important properties that the learner should possess.

**Flexibility of** $f$  The intent in few-shot segmentation is to be able to segment a wide range of classes, unseen during training. The image feature distributions of different unseen classes are not necessarily linearly separable (Allen et al., 2019). Prototypical few-shot learners, which are essentially linear classifiers, would fail in such scenarios. Instead, we need a mechanism that can learn and represent more complex functions $f$.

**Scalability in support set size** $K$  An FSS method should be able to effectively leverage additional support samples and therefore achieve substantially better accuracy and robustness for larger support sets. However, many prior works show little to no benefit in the 5-shot setting compared to 1-shot. As shown by Li et al. (2021) and Boudiaf et al. (2021), it is crucial that new information is effectively incorporated into the model without *averaging* out useful cues.

**Uncertainty Modeling**  Since only a small number of support samples are available in FSS, the network needs to regularly handle unseen appearances in the query images. For instance, the query may include novel backgrounds, scenes, and objects. Since the function $f$ predicted by the learner is not expected to generalize to such unseen scenarios, the network should instead utilize neighboring predictions or learned priors. However, this can only be achieved if $f$ models and communicates the uncertainty of its prediction to the decoder.

## 3.3 DENSE GAUSSIAN PROCESS FEW-SHOT LEARNER

In this work, we propose a dense Gaussian Process (GP) learner for few-shot segmentation. As a few-shot learning component, the GP possesses all three properties identified in the previous section. It can represent flexible and highly non-linear functions by selecting an appropriate kernel. As a non-parametric model, it further effectively benefits from additional support samples since all given data is retained. The GP explicitly models the uncertainty in the prediction by learning a probabilistic distribution over a space of functions. Furthermore, both the learning and inference steps of the GP are differentiable functions. While computational cost is a well-known challenge when deploying GPs, we demonstrate that even simple strategies can be used to keep the number of training samples at a tractable level for the FSS problem.

Our dense GP learner predicts a distribution of functions $f$ from the input features $x$ to an output $y$. Let $\mathbf{x}_{\mathcal{S}} \in \mathbb{R}^{KHW \times D}$ denote the matrix of all support image features and $\mathbf{x}_{\mathcal{Q}} \in \mathbb{R}^{HW \times D}$ the matrix retaining the query features. Let $\mathbf{y}_{\mathcal{S}}$ and $\mathbf{y}_{\mathcal{Q}}$ be the corresponding outputs. The former is obtained from the support masks and the latter is to be predicted with the GP. The key assumption

in the Gaussian process is that the support and query outputs $\mathbf{y}_\mathcal{S}, \mathbf{y}_\mathcal{Q}$ are jointly Gaussian according to,

$$\begin{pmatrix} \mathbf{y}_\mathcal{S} \\ \mathbf{y}_\mathcal{Q} \end{pmatrix} \sim \mathcal{N}\left( \begin{pmatrix} \boldsymbol{\mu}_\mathcal{S} \\ \boldsymbol{\mu}_\mathcal{Q} \end{pmatrix}, \begin{pmatrix} \mathbf{K}_{\mathcal{SS}} & \mathbf{K}_{\mathcal{SQ}} \\ \mathbf{K}_{\mathcal{SQ}}^\top & \mathbf{K}_{\mathcal{QQ}} \end{pmatrix} \right) \ . \tag{3}$$

For simplicity, we set the output prior means, $\boldsymbol{\mu}_\mathcal{S}$ and $\boldsymbol{\mu}_\mathcal{Q}$, to zero. The covariance matrix $\boldsymbol{K}$ in equation 3 is defined by the input features at those points and a *kernel* $\kappa : \mathbb{R}^D \times \mathbb{R}^D \to \mathbb{R}$. In our experiments, we adopt the commonly used *squared exponential* (SE) kernel

$$\kappa(x^m, x^n) = \sigma_f^2 \exp(-\frac{1}{2\ell^2}\|x^m - x^n\|_2^2) \ , \tag{4}$$

with scale parameter $\sigma_f$ and length parameter $\ell$. The kernel can be viewed as a similarity measure. If two features are similar, then the corresponding outputs are correlated.

Next, the posterior probability distribution of the query outputs is inferred. The rules for conditioning in a joint Gaussian gives us (Rasmussen & Williams, 2006)

$$\mathbf{y}_\mathcal{Q}|\mathbf{y}_\mathcal{S}, \mathbf{x}_\mathcal{S}, \mathbf{x}_\mathcal{Q} \sim \mathcal{N}(\boldsymbol{\mu}_{\mathcal{Q}|\mathcal{S}}, \boldsymbol{\Sigma}_{\mathcal{Q}|\mathcal{S}}) \ , \tag{5}$$

where

$$\boldsymbol{\mu}_{\mathcal{Q}|\mathcal{S}} = \mathbf{K}_{\mathcal{SQ}}^\top (\mathbf{K}_{\mathcal{SS}} + \sigma_y^2 \mathbf{I})^{-1} \mathbf{y}_\mathcal{S} \ , \tag{6}$$

$$\boldsymbol{\Sigma}_{\mathcal{Q}|\mathcal{S}} = \mathbf{K}_{\mathcal{QQ}} - \mathbf{K}_{\mathcal{SQ}}^\top (\mathbf{K}_{\mathcal{SS}} + \sigma_y^2 \mathbf{I})^{-1} \mathbf{K}_{\mathcal{SQ}} \ . \tag{7}$$

The measurements $\mathbf{y}_\mathcal{S}$ are assumed to have been obtained with some additive i.i.d. Gaussian noise with variance $\sigma_y^2$. This corresponds to adding a scaled identity matrix $\sigma_y^2 \mathbf{I}$ to the support covariance matrix $\mathbf{K}_{\mathcal{SS}}$. The equations 4-7 thus predict the distribution of the query outputs, represented by the mean value and the covariance. Note that the asymptotic complexity is $\mathcal{O}((KWH)^3)$. We therefore need to work on a sufficiently low resolution. Next, we introduce a way to incorporate more information into the predicted outputs while maintaining a low resolution.

### 3.4 LEARNING THE GP OUTPUT SPACE

The decoder strives to transform the query outputs predicted by the few-shot learner into an accurate mask. This is a challenging task given the low resolution and the desire to generalize to classes unseen during offline training. We therefore explore whether additional information can be encoded into the outputs, $\mathbf{y}_\mathcal{Q}$, in order to guide the decoder. This information could for instance include the shape of the mask or which of the object's parts is at a given location. To this end, we train a mask-encoder to construct the outputs,

$$\mathbf{y}_{\mathcal{S}k} = G(M_{\mathcal{S}k}) \ . \tag{8}$$

Here, $G$ is a neural network with learnable parameters.

The aforementioned formulation allows us to learn the GP output space during the meta-training stage. To the best of our knowledge, this has not previously been explored in the context of GPs. There is some reminiscence to the attention mechanism in transformers (Vaswani et al., 2017). The transformer queries, keys, and values correspond to our query features $\mathbf{x}_\mathcal{Q}$, support features $\mathbf{x}_\mathcal{S}$, and support outputs $\mathbf{y}_\mathcal{S}$. A difference is that in few-shot segmentation, there is a distinction between the features used for matching and the output - the features stem from the image whereas the output stems from the mask.

As mask encoder $G$, we employ a lightweight residual convolutional neural network. The network predicts multi-dimensional outputs for the support masks that are then reshaped into $\mathbf{y}_\mathcal{S} \in \mathbb{R}^{KHW \times E}$. These are fed into equation 6 when the posterior probability distribution of the query outputs is inferred. The two matrix-vector multiplications are transformed into matrix-matrix multiplications and the result $\boldsymbol{\mu}_{\mathcal{Q}|\mathcal{S}} \in \mathbb{R}^{HW \times E}$ is a matrix containing the multi-dimensional mean. The covariance in equation 7 is kept unchanged. This setup corresponds to the assumption that the covariance is isotropic over the output feature channels and that the different output feature channels are independent.

### 3.5 Final Mask Prediction

Next, we decode the predicted distributions of the query outputs in order to predict the final mask. Since the output of the GP is richer, including uncertainty and covariance information, we first need to consider what information to give to the final decoder network. We employ the following output features from the GP.

**GP output mean** The multi-dimensional mean $\boldsymbol{\mu}_{\mathcal{Q}|\mathcal{S}}$ represents the best guess of $\mathbf{y}_{\mathcal{Q}}$. Ideally, it contains a representation of the mask $M_{\mathcal{Q}}$ to be predicted. The decoder works on 2D feature maps and $\boldsymbol{\mu}_{\mathcal{Q}|\mathcal{S}}$ is therefore reshaped as $\mathbf{z}_\mu \in \mathbb{R}^{H \times W \times E}$.

**GP output covariance** The covariance $\boldsymbol{\Sigma}_{\mathcal{Q}|\mathcal{S}}$ captures both the uncertainty in the predicted query output $y_{\mathcal{Q}}$ and the correlation between different query outputs. The former lets the decoder network to identify uncertain regions in the image, and instead rely on, e.g., learnt priors or neighbouring predictions. Moreover, local correlations can tell whether two locations of the image are similar. For each query output, we employ the covariance between it and each of its spatial neighbours in an $N \times N$ region. We represent it as a 2D feature map $\mathbf{z}_\Sigma \in \mathbb{R}^{H \times W \times (N^2)}$ by vectorizing the $N^2$ covariance values in the $N \times N$ neighborhood. For more details, see Appendix E.

**Shallow image features** Shallow image features, extracted from early layers of deep neural networks, are of high resolution and permit precise localization of object boundaries (Bhat et al., 2018). We therefore store feature maps extracted from earlier layers of $F$ and feed them into the decoder. These serve to guide the decoder as it transforms the low-resolution output mean into a precise, high-resolution mask.

We finally predict the query mask by feeding the above information into a decoder $U$,

$$\widehat{M}_{\mathcal{Q}} = U(\mathbf{z}_\mu, \mathbf{z}_\sigma, \mathbf{z}_\Sigma, \mathbf{x}_{\mathcal{Q},\text{shallow}}) \ . \tag{9}$$

As a decoder $U$, we adopt DFN (Yu et al., 2018). DFN processes its input at one scale at a time, starting with the coarsest scale. In our case, this is the mean and covariance. At each scale, the result at the previous scale is upsampled and processed together with any input at that scale. After processing the finest scaled input, the result is upsampled to the mask scale and classified with a linear layer and a `softmax` function.

### 3.6 FSS Learner Pyramid

Many computer vision tasks benefit from processing at multiple scales or multiple feature levels. While deep, high-level features directly capture the presence of objects or semantic classes, the mid-level features capture object parts Tian et al. (2020). In semantic segmentation, methods begin with high-level features and then successively adds mid-level features while upsampling (Long et al., 2015). In object detection, a detection head is applied to each level in a feature pyramid (Lin et al., 2017). Tian et al. (2020) discusses and demonstrates the benefit of using both mid-level and high-level features for few-shot segmentation. We therefore adapt our framework to be able to process features at different levels.

We take features from our image encoder $F$ extracted at multiple levels $A$. Let the support and query features at level $a \in A$ be denoted as $\mathbf{x}_{\mathcal{S}}^a$ and $\mathbf{x}_{\mathcal{Q}}^a$. From the mask encoder $G$ we extract corresponding support outputs $\mathbf{y}_{\mathcal{S}}^a$. We then introduce a few-shot learner $\Lambda^a$ for each level $a$. For efficient inference, the support features and outputs are sampled on a grid such that the total stride compared to the original image is 32. The query features retain their resolution. Each few-shot learner then infers a posterior distribution over the query outputs $\mathbf{y}_{\mathcal{Q}}^a$, parameterized by $\boldsymbol{\mu}_{\mathcal{Q}|\mathcal{S}}^a$ and $\boldsymbol{\Sigma}_{\mathcal{Q}|\mathcal{S}}^a$. These are then fed into the decoder, which processes them one scale at a time.

## 4 Experiments

We validate our proposed approach by conducting comprehensive experiments on two FSS benchmarks: PASCAL-$5^i$ (Shaban et al., 2017) and COCO-$20^i$ (Nguyen & Todorovic, 2019).

### 4.1 EXPERIMENTAL SETUP

**Datasets** We conduct experiments on the PASCAL-$5^i$ (Shaban et al., 2017) and COCO-$20^i$ (Nguyen & Todorovic, 2019) benchmarks. PASCAL-$5^i$ is composed of PASCAL VOC 2012 (Everingham et al., 2010) with additional SBD (Hariharan et al., 2011) annotations. The dataset comprises 20 categories split into 4 folds. For each fold, 15 categories are used for training and the remaining 5 for testing. COCO-$20^i$ (Nguyen & Todorovic, 2019) is more challenging and is built from MS-COCO (Lin et al., 2014). Similar to PASCAL-$5^i$, COCO-$20^i$ benchmark is split into 4 folds. For each fold, 60 base classes are used for training and the remaining 20 for testing.

**Implementation Details** Following previous works, we employ ResNet-50 and ResNet-101 (He et al., 2016) backbones pre-trained on ImageNet (Russakovsky et al., 2015) as image encoders. We let the dense GP work with feature maps produced by the third and the fourth residual module. In addition, we place a single convolutional projection layer that reduces the feature map down to 512 dimensions. As mask encoder, we use a light-weight CNN (see Appendix E). We use $\sigma_y^2 = 0.1$, $\sigma_f^2 = 1$, and $\ell^2 = \sqrt{D}$ in our GP. We train our models for 20k and 40k iterations à 8 episodes for PASCAL-$5^i$ and COCO-$20^i$, respectively. On one NVIDIA A100 GPU, this takes up to 10 hours. We use the AdamW (Kingma & Ba, 2015; Loshchilov & Hutter, 2019) optimizer with a weight decay factor of $10^{-3}$ and a cross-entropy loss with 1-4 background-foreground weighting. We use a learning rate of $5 \cdot 10^{-5}$ for all parameters save for the image encoder, which uses a learning rate of $10^{-6}$. The learning rate is decayed with a factor of 0.1 when 10k iterations remain. We freeze the batch normalization layers of the image encoder. Episodes are sampled in the same way as we evaluate. We randomly flip images horizontally followed by resizing to a size of $384 \times 384$ for PASCAL-$5^i$ and $512 \times 512$ for COCO-$20^i$. Well-documented code will be made publicly available.

**Evaluation** We evaluate our approach on each fold by randomly sampling 5k and 20k episodes respectively, for PASCAL-$5^i$ and COCO-$20^i$. This follows the work of Tian et al. (2020) in which it is observed that the procedure employed by many prior works, using only 1000 episodes, yields fairly high variance. Additionally, following Wang et al. (2019); Liu et al. (2020c); Boudiaf et al. (2021); Zhang et al. (2021), our results in the state-of-the-art comparison are computed as the average of 5 runs with different seeds. We also report the standard deviation. Performance is measured in terms of mean Intersection over Union (mIoU). First, the IoU is calculated per class over all episodes in a fold. The mIoU is then obtained by averaging the IoU over the classes. As in the original work on FSS by Shaban et al. (2017), we calculate the IoU on the original resolution of the images.

### 4.2 STATE-OF-THE-ART COMPARISON

We compare our proposed DGPNet with existing FSS approaches on PASCAL-$5^i$ and COCO-$20^i$ benchmarks. Table 1 presents the state-of-the-art comparison on both benchmarks. Following prior works, we report results given a single support example, *1-shot*, and given five support examples, *5-shot*. Our proposed DGPNet sets a new state-of-the-art on *both* benchmarks. Among the previous approaches, SAGNN (Xie et al., 2021) achieves mIoU scores of 62.1 and 62.8 for 1-shot and 5-shot segmentation, respectively on PASCAL-$5^i$ using ResNet50. RePri (Boudiaf et al., 2021) obtains mIoU scores of 59.1 and 66.8 for 1-shot and 5-shot segmentation, respectively. Our DGPNet achieves superior performance with mIoU scores of 63.5 and 73.5 for 1-shot and 5-shot segmentation, respectively. When using the ResNet101 backbone, PFENet (Tian et al., 2020) achieves the best 1-shot segmentation performance of 60.1 among existing methods. Our DGPNet outperforms PFENet (Tian et al., 2020) by achieving mIoU score of 64.8. Similarly, our DGPNet significantly outperforms the best reported 5-shot results in literature (65.6 by RePri (Boudiaf et al., 2021)), with an absolute gain of 9.8.

| Support Set | Query | DGPNet | Support Set | Query | DGPNet |

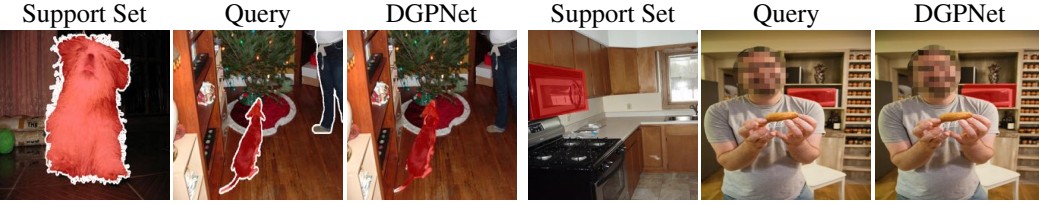

Figure 3: Qualitative 1-shot results of our approach on PASCAL-$5^i$ (left) and COCO-$20^i$ (right).

Table 1: State-of-the-art comparison on PASCAL-$5^i$ and COCO-$20^i$ benchmark in terms of mIoU (higher is better). In each case, the best two results are shown in magenta and cyan font, respectively. Asterisk $^*$ denotes re-implementation by Liu et al. (2020a). Our DGPNet achieves state-of-the-art results for 1-shot and 5-shot on both benchmarks. When using ResNet101, our DGPNet achieves absolute gains of 8.2 and 15.2 for 1-shot and 5-shot segmentation, respectively on the challenging COCO-$20^i$ benchmark, compared to the best reported results in the literature.

| Backbone | Method | PASCAL-$5^i$ | | COCO-$20^i$ | |
|---|---|---|---|---|---|
| | | 1-shot | 5-shot | 1-shot | 5-shot |
| ResNet50 | CANet (Zhang et al., 2019b) | 55.4 | 57.1 | 40.8* | 42.0* |
| | PGNet (Zhang et al., 2019a) | 56.0 | 58.5 | 36.7* | 37.5* |
| | RPMM (Yang et al., 2020a) | 56.3 | 57.3 | 30.6 | 35.5 |
| | CRNet (Liu et al., 2020b) | 55.7 | 58.8 | - | - |
| | DENet (Liu et al., 2020a) | 60.1 | 60.5 | *42.8* | *43.0* |
| | LTM (Yang et al., 2020b) | 57.0 | 60.6 | - | - |
| | PFENet (Tian et al., 2020) | 60.8 | 61.9 | - | - |
| | PPNet (Liu et al., 2020c) | 51.5 | 62.0 | 25.7 | 36.2 |
| | RePri (Boudiaf et al., 2021) | 59.1 | *66.8* | 34.0 | 42.1 |
| | SCL (Zhang et al., 2021) | 61.8 | 62.9 | - | - |
| | SAGNN (Xie et al., 2021) | *62.1* | 62.8 | - | - |
| | **DGPNet (Ours)** | **63.5 ± 0.4** | **73.5 ± 0.3** | **45.0 ± 0.4** | **56.2 ± 0.4** |
| ResNet101 | FWB (Nguyen & Todorovic, 2019) | 56.2 | 60.0 | 21.2 | 23.7 |
| | DAN (Wang et al., 2020) | 58.2 | 60.5 | 24.4 | 29.6 |
| | PFENet (Tian et al., 2020) | *60.1* | 61.4 | *38.5* | *42.7* |
| | RePri (Boudiaf et al., 2021) | 59.4 | *65.6* | - | - |
| | VPI (Wang et al., 2021) | 57.3 | 60.4 | 23.4 | 27.8 |
| | ASGNet (Li et al., 2021) | 59.3 | 64.4 | 34.6 | 42.5 |
| | SCL (Zhang et al., 2021) | - | - | 37.0 | 39.9 |
| | SAGNN (Xie et al., 2021) | - | - | 37.2 | *42.7* |
| | **DGPNet (Ours)** | **64.8 ± 0.5** | **75.4 ± 0.4** | **46.7 ± 0.3** | **57.9 ± 0.3** |

On the challenging COCO-$20^i$ benchmark, DENet (Liu et al., 2020a) obtains mIoU scores of 42.8 and 43.0 for 1-shot and 5-shot segmentation, respectively using ReNet50. Our DGPNet outperforms DENet (Liu et al., 2020a), achieving mIoU scores of 45.0 and 56.2 for 1-shot and 5-shot segmentation, respectively. When using ResNet101, DGPNet obtains absolute gains of 8.2 and 15.2 for 1-shot and 5-shot segmentation, respectively, compared to the best reported results in literature. Our per-fold results, including the standard deviation for each fold, are provided in appendix A.3. Qualitative examples are found in Figure 3 and Appendix F.

**Scaling Segmentation Performance with Increased Support Size** As discussed earlier, the FSS approach is desired to effectively leverage additional support samples, thereby achieving superior accuracy and robustness for larger shot-sizes. We therefore analyze the effectiveness of the proposed DGPNet by increasing the number of shots on PASCAL-$5^i$ and COCO-$20^i$. Figure 1 shows the results on the two benchmarks. We use the model trained for 5-shot for in all cases, except for 1-shot. Compared to most existing works, our DGPNet effectively benefits from additional support samples, achieving remarkable improvement in segmentation accuracy with larger support sizes.

**Cross-dataset Evaluation** In Table 2, we present the cross-dataset evaluation capabilities of our DGPNet from COCO-$20^i$ to PASCAL. For this cross-dataset evaluation experiment, we followed the same protocol as in Boudiaf et al. (2021), where the folds are constructed to ensure that there is no overlap with COCO-$20^i$ training folds. When using the same ResNet50 backbone, our approach obtains 1-shot and 5-shot mIoU of 68.9 and 77.5, respectively with absolute gains of 5.8 and 11.3 over RePRI (Boudiaf et al., 2021). Further, DGPNet achieves segmentation mIoU of 70.1 and 78.5 in 1-shot and 5-shot setting, respectively using ResNet101. For more details on the experiment, see Appendix A.1.

Table 2: Performance comparison (mIoU, higher is better), when performing cross-dataset evaluation from COCO-$20^i$ to PASCAL. When using the same ResNet50 backbone, our approach achieves significant improvements for both 1-shot and 5-shot settings, with absolute gains of 5.8 and 11.3 mIoU over RePRI (Boudiaf et al., 2021).

| Method | 1-Shot | 5-Shot |
|---|---|---|
| RPMM (Yang et al., 2020a) | 49.6 | 53.8 |
| PFENet (Tian et al., 2020) | 61.1 | 63.4 |
| RePRI (Boudiaf et al., 2021) | *63.1* | *66.2* |
| Ours, ResNet50 | **68.9 ± 0.4** | **77.5 ± 0.2** |
| Ours, ResNet101 | **70.1 ± 0.3** | **78.5 ± 0.3** |

### 4.3 ABLATION STUDY

Here, we analyze the impact of the key components in our proposed architecture. We first investigate the choice of kernel, $\kappa$. Then, we analyze the impact of the predictive covariance provided by the GP and the benefits of learning the GP output space. Last, we experiment with dense GPs at multiple feature levels. All experiments in this section are conducted with the ResNet50 backbone. Detailed experiments are provided in the Appendix.

**Choice of Kernel** Going from a linear few-shot learner to a more flexible function requires an appropriate choice of kernel. We consider the *homogenous linear kernel* as our baseline. Note that the homogenous linear kernel is equivalent to Bayesian linear regression under appropriate priors (Rasmussen & Williams, 2006). To make our learner more flexible, we consider two additional choices of kernels, the *Exponential* and *Squared Exponential* (SE) kernels. The kernel equations are given in Appendix C. In Table 3, results on both the PASCAL and COCO benchmarks are presented. Both the Exponential and SE kernels greatly outperform the linear kernel, with the SE kernel leading to a significant gain of 5.5 in mIoU. These results show the benefit of a more flexible and scalable learner.

Table 3: Performance of different kernels on the PASCAL-$5^i$ and COCO-$20^i$ benchmarks. Notably, both the Exponential and SE kernels significantly outperform the linear kernel, confirming the need for a flexible learner. Measured in mIoU (higher is better). Best results are in bold.

| Kernel | PASCAL-$5^i$ | | COCO-$20^i$ | | |
| --- | --- | --- | --- | --- | --- |
| | 1-shot | 5-shot | 1-shot | 5-shot | $\Delta$ |
| Linear | 58.6 | 61.8 | 37.1 | 45.3 | 0.0 |
| Exponential | 59.3 | 67.3 | 40.9 | **51.8** | 4.1 |
| SE | **62.1** | **69.9** | **41.7** | 51.2 | **5.5** |

**Incorporating Uncertainty and Learning the GP Output Space** We adopt the SE kernel and analyze the performance of letting the decoder process the predictive covariance provided by the dense GP. With the covariance in a $5 \times 5$ window, we obtain a gain of 1.7 mIoU. Then, we add the mask-encoder and learn the GP output space (GPO). This leads to a 1.7 improvement in isolation, or a 2.5 mIoU gain together with the covariance.

Table 4: Analysis of learning the GP output space (GPO) and incorporating covariance (Cov). Performance in mIoU (higher is better). Best results are in bold.

| Cov | GPO | PASCAL-$5^i$ | | COCO-$20^i$ | | |
| --- | --- | --- | --- | --- | --- | --- |
| | | 1-shot | 5-shot | 1-shot | 5-shot | $\Delta$ |
| | | 62.1 | 69.9 | 41.7 | 51.2 | 0.0 |
| ✓ | | **62.5** | 71.8 | 43.8 | 53.7 | 1.7 |
| | ✓ | 61.7 | **72.7** | 43.1 | 54.2 | 1.7 |
| ✓ | ✓ | **62.5** | 72.6 | **44.7** | **55.0** | **2.5** |

**Multilevel Representations** Finally, we investigate the effect of introducing a multilevel hierarchy of dense GPs. Specifically we investigate using combinations of stride 16 and 32 features. The results are reported in Table 5. The results show the benefit of using dense GPs at different feature levels, leading to an average gains of 1.4 mIoU.

Table 5: Performance of different multilevel configurations of the dense GP few-shot learner on the PASCAL-$5^i$ and COCO-$20^i$ benchmarks. Measured in mIoU (higher is better). Best results are in bold.

| Stride 16 | Stride 32 | PASCAL-$5^i$ | | COCO-$20^i$ | | |
| --- | --- | --- | --- | --- | --- | --- |
| | | 1-shot | 5-shot | 1-shot | 5-shot | $\Delta$ |
| | ✓ | 62.5 | 72.6 | 44.7 | 55.0 | 0.0 |
| ✓ | | 60.4 | 69.3 | 40.7 | 51.1 | -3.9 |
| ✓ | ✓ | **63.9** | **73.6** | **45.3** | **56.4** | **1.4** |

## 5 CONCLUSION

We have proposed a few-shot learner based on Gaussian process regression for the few-shot segmentation task. The GP models the support set in deep feature space and its flexibility permits it to capture complex feature distributions. It makes probabilistic predictions on the query image, providing both a point estimate and additional uncertainty information. These predictions are fed into a CNN decoder that predicts the final segmentation. The resulting approach sets a new state-of-the-art on PASCAL-$5^i$ and COCO-$20^i$, with absolute improvements of up to 14.9 mIoU. Our approach scales well with larger support sets during inference, even when trained for a fixed number of shots.

**Reproducibility Statement** In addition to the data, implementation, and evaluation details provided in section 4.1, we also provide (i) pseudo-code for the Gaussian Process; (ii) the details of the input to the decoder, and (iii) a table including all trainable layers of our network. We have tried to be complete and believe that this information is sufficient to reproduce our results. In addition, the code for both training and evaluation will be made publicly available. Furthermore, due to the stochastic nature of the task, we follow Tian et al. (2020) and try to reduce the latter by sampling more episodes, 5000 for PASCAL-$5^i$ and 20000 for COCO-$20^i$. We additionally train and evaluate our final approach with five different seeds. Both the performance mean and standard deviation are reported in Table 1.

**Ethics Statement** Computer vision has the potential to automate many tasks, both in the form of autonomous agents and in the form of big-data processing. There is a wide range of potential benefits in doing so, such as enabling robots to perform hazardous tasks or via advanced driving assistance systems reduce the number of accidents in traffic. However, there are also challenges. Automation may require changes to societal systems and can be used for ill. We do not perceive few-shot segmentation as particularly problematic. Instead, few-shot segmentation enables the introduction of new categories without the need of large-scale data collection. Approaches based on episodic training, including our approach, are instead trained only on standard computer vision datasets. We see two potential benefits. First, it democratizes semantic segmentation as the cost of data collection is reduced. Secondly, with large-scale data collection there is always a risk of privacy infringement. Reducing the amount of data necessary mitigates these risks.

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

## SUPPLEMENTARY MATERIAL

We first provide additional results in Appendix A. Then, in Appendix B we provide analyses of the covariance neighborhood size and the effect of training the image encoder. The equations for the different kernels used in our ablation study are shown in Appendix C. In Appendix D, we list the runtimes of the different components in our approach. Appendix E contains pseudo-code for our dense GP, a list detailing the trainable layers of our approach, and the details on how the output of the GP is presented to the decoder. Finally, in Appendix F, we provide a qualitative comparison and qualitative results.

## A   ADDITIONAL RESULTS

We provide the per-fold results on the COCO-$20^i$ to PASCAL transfer experiment and a list of the classes included; the per-shot results used to generate Figure 1 in the main paper; and the per-fold results used in the state-of-the-art comparison.

### A.1   CROSS-DATASET EVALUATION

We supply additional details on the cross-dataset evaluation experiment. This experiment was proposed by Boudiaf et al. (2021) and we follow their setup. First, our approach is trained on each of the four folds of COCO-$20^i$. Next, we test each of the four versions on PASCAL, using only the classes held-out during training. We list the classes of each fold in Table 7. The full per-fold results are presented in Table 6.

Table 6: The results of our approach in a COCO-$20^i$ to PASCAL transfer experiment (mIoU, higher is better). Following Boudiaf et al. (2021), the approach is trained on a fold of COCO-$20^i$ training set and tested on the PASCAL validation set. The testing folds are constructed to include classes not present in the training set, and thus not the same as PASCAL-$5^i$.

| Method | 1-Shot | | | | | 5-Shot | | | | |
|---|---|---|---|---|---|---|---|---|---|---|
| | F-0 | F-1 | F-2 | F-3 | Mean | F-0 | F-1 | F-2 | F-3 | Mean |
| RPMM (Yang et al., 2020a) | 36.3 | 55.0 | 52.5 | 54.6 | 49.6 | 40.2 | 58.0 | 55.2 | 61.8 | 53.8 |
| PFENet (Tian et al., 2020) | 43.2 | 65.1 | 66.5 | 69.7 | 61.1 | 45.1 | 66.8 | 68.5 | 73.1 | 63.4 |
| RePRI (Boudiaf et al., 2021) | 52.8 | 64.0 | 64.1 | 71.5 | 63.1 | 57.7 | 66.1 | 67.6 | 73.1 | 66.2 |
| Ours, ResNet50 | 55.1 ± 0.8 | 71.0 ± 0.4 | 69.2 ± 0.9 | 80.3 ± 0.8 | 68.9 ± 0.4 | 70.3 ± 0.9 | 75.3 ± 0.4 | 78.5 ± 0.7 | 85.8 ± 0.4 | 77.5 ± 0.2 |
| Ours, ResNet101 | 55.1 ± 0.4 | 72.2 ± 0.3 | 70.7 ± 0.8 | 82.3 ± 0.8 | 70.1 ± 0.3 | 70.7 ± 0.7 | 75.6 ± 0.6 | 80.2 ± 0.1 | 87.3 ± 0.3 | 78.5 ± 0.3 |

Table 7: The classes used for testing in the COCO-$20^i$ to PASCAL transfer experiment, as proposed by Boudiaf et al. (2021). This split is different from that of PASCAL-$5^i$ in order to avoid overlap between the training and testing classes.

| Fold-0 | Fold-1 | Fold-2 | Fold-3 |
|---|---|---|---|
| Airplane, Boat, Chair, Dining Table, Dog, Person | Bicycle, Bus, Horse, Sofa | Bird, Car, Potted Plant, Sheep, Train, TV-monitor | Bottle, Cat, Cow, Motorcycle |

### A.2   1-10 SHOT RESULTS

Here, we provide the full results from 1-10 shots, which was used to generate Figure 1. In Table 8 and 9 our results on PASCAL-$5^i$ and COCO-$20^i$ are presented. Note that our approach was trained for 1 shot in the 1-shot setting, and 5 shots in the other nine settings.

Table 8: 1 to 10 -shot PASCAL-$5^i$ results.

| | 1-shot | 2-shot | 3-shot | 4-shot | 5-shot | 6-shot | 7-shot | 8-shot | 9-shot | 10-shot |
|---|---|---|---|---|---|---|---|---|---|---|
| DGPNet (ResNet101) | 64.8 ± 0.5 | 68.4 ± 0.5 | 72.4 ± 0.3 | 74.2 ± 0.5 | 75.4 ± 0.4 | 76.1 ± 0.4 | 76.6 ± 0.4 | 77.0 ± 0.4 | 77.5 ± 0.4 | 77.7 ± 0.4 |

Table 9: 1 to 10 -shot COCO-$20^i$ results.

| | 1-shot | 2-shot | 3-shot | 4-shot | 5-shot | 6-shot | 7-shot | 8-shot | 9-shot | 10-shot |
|---|---|---|---|---|---|---|---|---|---|---|
| DGPNet (ResNet101) | 46.8 ± 0.3 | 51.7 ± 0.2 | 55.0 ± 0.2 | 56.7 ± 0.3 | 57.9 ± 0.3 | 58.7 ± 0.2 | 59.2 ± 0.3 | 59.7 ± 0.3 | 60.0 ± 0.3 | 60.2 ± 0.2 |

### A.3 Per-Fold Results

In this section, we provide the full results for our state-of-the-art comparison. In Tables 10 and 11 the per-fold results for both ResNet50 and ResNet101 are presented. In general our results are stable for all folds.

Table 10: Per-fold results on PASCAL-$5^i$

| Method | 1-Shot | | | | | 5-Shot | | | | |
|---|---|---|---|---|---|---|---|---|---|---|
| | $5^0$ | $5^1$ | $5^2$ | $5^3$ | Mean | $5^0$ | $5^1$ | $5^2$ | $5^3$ | Mean |
| CANet (ResNet50) | 52.5 | 65.9 | 51.3 | 51.9 | 55.4 | 55.5 | 67.8 | 51.9 | 53.2 | 57.1 |
| DENet (ResNet50) | 55.7 | 69.7 | 63.6 | 51.3 | 60.1 | 54.7 | 71.0 | 64.5 | 51.6 | 60.5 |
| PFENet (ResNet50) | 61.7 | 69.5 | 55.4 | 56.3 | 60.8 | 63.1 | 70.7 | 55.8 | 57.9 | 61.9 |
| RePri (ResNet50) | 60.2 | 67.0 | 61.7 | 47.5 | 59.1 | 64.5 | 70.8 | 71.7 | 60.3 | 66.8 |
| ASGNet (ResNet101) | 59.8 | 67.4 | 55.6 | 54.4 | 59.3 | 64.6 | 71.3 | 64.2 | 57.3 | 64.4 |
| SCL (ResNet50) | 63.0 | 70.0 | 56.5 | 57.7 | 61.8 | 64.5 | 70.9 | 57.3 | 58.7 | 62.9 |
| SAGNN (ResNet50) | 64.7 | 69.6 | 57.0 | 57.2 | 62.1 | 64.9 | 70.0 | 57.0 | 59.3 | 62.8 |
| DGPNet (ResNet50) | $63.5 \pm 0.9$ | $71.1 \pm 0.5$ | $58.2 \pm 1.6$ | $61.2 \pm 0.7$ | $63.5 \pm 0.4$ | $72.4 \pm 0.8$ | $76.9 \pm 0.2$ | $73.2 \pm 0.9$ | $71.7 \pm 0.4$ | $73.5 \pm 0.3$ |
| DGPNet (ResNet101) | $63.9 \pm 1.2$ | $71.0 \pm 0.5$ | $63.0 \pm 0.6$ | $61.4 \pm 0.6$ | $64.8 \pm 0.5$ | $74.1 \pm 0.6$ | $77.4 \pm 0.6$ | $76.7 \pm 0.9$ | $73.4 \pm 0.6$ | $75.4 \pm 0.4$ |

Table 11: Per-fold results on COCO-$20^i$

| Method | 1-Shot | | | | | 5-Shot | | | | |
|---|---|---|---|---|---|---|---|---|---|---|
| | $20^0$ | $20^1$ | $20^2$ | $20^3$ | Mean | $20^0$ | $20^1$ | $20^2$ | $20^3$ | Mean |
| DENet (ResNet50) | 42.9 | 45.8 | 42.2 | 40.2 | 42.8 | 45.4 | 44.9 | 41.6 | 40.3 | 43.0 |
| PFENet (ResNet50) | 36.8 | 41.8 | 38.7 | 36.7 | 38.5 | 40.4 | 46.8 | 43.2 | 40.5 | 42.7 |
| RePri (ResNet50) | 31.2 | 38.1 | 33.3 | 33.0 | 34.0 | 38.5 | 46.2 | 40.0 | 43.6 | 42.1 |
| ASGNet (ResNet50) | - | - | - | - | 34.6 | - | - | - | - | 42.5 |
| SCL (ResNet101) | 36.4 | 38.6 | 37.5 | 35.4 | 37.0 | 38.9 | 40.5 | 41.5 | 38.7 | 39.9 |
| SAGNN (ResNet101) | 36.1 | 41.0 | 38.2 | 33.5 | 37.2 | 40.9 | 48.3 | 42.6 | 38.9 | 42.7 |
| DGPNet (ResNet50) | $43.6 \pm 0.5$ | $47.8 \pm 0.8$ | $44.5 \pm 0.8$ | $44.2 \pm 0.6$ | $45.0 \pm 0.4$ | $54.7 \pm 0.7$ | $59.1 \pm 0.5$ | $56.8 \pm 0.6$ | $54.4 \pm 0.6$ | $56.2 \pm 0.4$ |
| DGPNet (ResNet101) | $45.1 \pm 0.5$ | $49.5 \pm 0.8$ | $46.6 \pm 0.8$ | $45.6 \pm 0.3$ | $46.7 \pm 0.3$ | $56.8 \pm 0.8$ | $60.4 \pm 0.9$ | $58.4 \pm 0.4$ | $55.9 \pm 0.4$ | $57.9 \pm 0.3$ |

## B  Detailed Analysis

We present results from two additional experiments. First, the size of the local covariance region is analyzed. Then, we investigate the impact of freezing the backbone during episodic training.

### B.1  Additional Covariance Neighborhood Experiments

In section 3.5 we show how the covariance with neighbors in a local region is fed to the decoder. In Table 12, we supply additional results with different sized windows. We experiment with $N \in \{1, 3, 5, 7\}$. Larger windows improve the results but the improvement seems to saturate after $N = 5$.

Table 12: Performance for different configurations of covariance windows on the PASCAL-$5^i$ and COCO-$20^i$ benchmarks. Measured in mIoU (higher is better).

| 1x1 | 3x3 | 5x5 | 7x7 | PASCAL-$5^i$ | | COCO-$20^i$ | | |
|---|---|---|---|---|---|---|---|---|
| | | | | 1-shot | 5-shot | 1-shot | 5-shot | Δ |
| | | | | 62.1 | 69.9 | 41.7 | 51.2 | 0.0 |
| ✓ | | | | 60.0 | 70.3 | 42.2 | 51.7 | -0.2 |
| | ✓ | | | 62.0 | 71.1 | 43.6 | 53.0 | 1.2 |
| | | ✓ | | 62.5 | **71.8** | **43.8** | **53.7** | **1.7** |
| | | | ✓ | **63.0** | 71.7 | 43.7 | 53.5 | **1.8** |

### B.2  Effect of Training the Image Encoder

In Table 13 we compare our final approach trained with a frozen and unfrozen backbone. Several prior works found it beneficial to freeze the image encoder during episodic training. In contrast, we find it beneficial to not freeze it and keep learning visual representations. By differentiating through our GP learner during episodic training, the proposed method can thus refine the underlying feature representations, which is another important advantage of our approach. However, our approach still improves upon state-of-the-art when employing a frozen backbone.

### B.3  Additional Baseline Experiments

We supply three additional baseline experiments to validate the effect of the proposed DGP-module. We evaluate three variants: (i) We remove the GP (the f-branch) and let the decoder rely only on the shallow features. (ii) We replace the GP with the Prior-mask learning mechanism proposed in

Table 13: Comparison between freezing the backbone and fine-tuning it with a low learning rate.

| Backbone | Method | PASCAL-$5^i$ | | COCO-$20^i$ | |
| | | 1-shot | 5-shot | 1-shot | 5-shot |
|---|---|---|---|---|---|
| ResNet50 | DGPNet (Frozen Backbone) | $61.9 \pm 0.3$ | $72.4 \pm 0.3$ | $43.1 \pm 0.3$ | $54.5 \pm 0.2$ |
| | **DGPNet (Ours)** | $\mathbf{63.5 \pm 0.4}$ | $\mathbf{73.5 \pm 0.3}$ | $\mathbf{45.0 \pm 0.4}$ | $\mathbf{56.2 \pm 0.4}$ |
| ResNet101 | DGPNet (Frozen Backbone) | $63.4 \pm 0.5$ | $74.3 \pm 0.3$ | $44.6 \pm 0.5$ | $56.6 \pm 0.3$ |
| | **DGPNet (Ours)** | $\mathbf{64.8 \pm 0.5}$ | $\mathbf{75.4 \pm 0.4}$ | $\mathbf{46.7 \pm 0.3}$ | $\mathbf{57.9 \pm 0.3}$ |

PFENet. (iii) We replace the GP with a prototype-based approach where the support features are mask-pooled and the result compared to the query features using cosine-similarity, similar to e.g. PANet. Results on PASCAL and COCO are reported in the Table 14. Our approach outperforms all baselines by a large margin. This further validates the superiority of our GP module.

Table 14: Additional baseline experiments to validate the effect of the DGP-module.

| Method | PASCAL-$5^i$ | | COCO-$20^i$ | |
| | 1-shot | 5-shot | 1-shot | 5-shot |
|---|---|---|---|---|
| No GP | 42.9 | 43.0 | 23.5 | 23.5 |
| PFENet prior mask | 54.0 | 54.7 | 35.8 | 37.2 |
| Prototype | 57.5 | 63.0 | 41.5 | 49.9 |
| **DGPNet (Ours)** | **63.5** | **73.5** | **45.0** | **56.2** |

## C KERNEL DETAILS

In this section we provide the definitions of the kernels used in our ablation study. First we define the homogenous linear kernel as,

$$\kappa_{\text{lin}}(x, y) = x^T y \ .$$

As was noted in the paper, this kernel corresponds to Bayesian Linear Regression with a specific prior. One could also consider learning a bias parameter, however we chose not to learn such parameters for reasons of method simplicity. Next we consider the exponential kernel,

$$\kappa_{\text{exp}}(x, y) = \exp\left( -\frac{||x - y||_2}{\ell} \right) .$$

This kernel behaves similarly as the SE kernel, however with a sharper peak and slower rate of decay. For completeness we additionally define the SE kernel here again,

$$\kappa_{\text{SE}}(x, y) = \exp\left( -\frac{||x - y||_2^2}{2\ell^2} \right) .$$

We chose the length of the exponential kernel as $\ell = \sqrt{D}$ and the squared exponential kernel as $\ell^2 = \sqrt{D}$, we found the performance in general to be robust to different values of $\ell$. Note that we used the same value of $\ell$ for both benchmarks and for all number of shots.

In Table 15 we provide a kernel hyperparameter sensitivity analysis. We run one experiment per value and choose the values to cover one order of magnitude centered around the values adopted. Perturbing the hyperparameters with a factor of $0.3$ or $3.0$ leads to minor but statistically significant drops in performance. One exception is increasing $\sigma_f^2$ by a factor of $3.0$, which does not adversely affect performance.

Table 15: Kernel hyperparameter sensitivity analysis for the SE-kernel, which is adopted in the main paper. The sensitivity analysis is based on the full, final approach with a ResNet50 backbone and performed on the 5-shot setting in PASCAL-$5^i$. The hyperparameters values are chosen to cover one order of magnitude of hyperparameter values, centered at the values adopted in the main paper.

| $\ell^2 = 0.3\sqrt{D}$ | $\ell^2 = \sqrt{D}$ | $\ell^2 = 3.0\sqrt{D}$ | $\sigma_f^2 = 0.3$ | $\sigma_f^2 = 1.0$ | $\sigma_f^2 = 3.0$ | $\sigma_y^2 = 0.03$ | $\sigma_y^2 = 0.1$ | $\sigma_y^2 = 0.3$ |
|---|---|---|---|---|---|---|---|---|
| 72.9 | **73.5$\pm$ 0.3** | 72.8 | 72.8 | **73.5$\pm$ 0.3** | 73.6 | 73.0 | **73.5$\pm$ 0.3** | 72.9 |

## D RUNTIMES

We show the runtime of our method in Table 16. We partition the timing into different parts. The Gaussian process (GP) is split into two. One part preparing and decomposing the support set matrix

Table 16: Runtimes of the different functions in our approach, measured in milliseconds (ms). Timings are measured in evaluation mode on $512 \times 512$ sized images from COCO-20$^i$.

| | Batch size 1 | | Batch size 20 | |
|---|---|---|---|---|
| | 1-shot | 5-shot | 1-shot | 5-shot |
| Image encoder on support | 15.5 | 23.4 | 48.5 | 197.5 |
| Mask encoder on support | 2.1 | 2.1 | 1.6 | 6.6 |
| GP preparation on support | 8.3 | 13.9 | 7.2 | 113.5 |
| Image encoder on query | 9.7 | 13.6 | 39.8 | 39.9 |
| GP inference on query | 7.9 | 9.4 | 12.1 | 38.1 |
| Decoder | 6.2 | 6.9 | 40.0 | 40.5 |
| Total | 49.7 ms | 69.3 ms | 149.2 ms | 436.1 ms |

in equation 6 and equation 7 of the main paper, and another part computing the mean and covariance of the query given the pre-computed matrix decomposition.

The timings are measured in the 1-shot and 5-shot settings on images from COCO-20$^i$ of $512 \times 512$ resolution, using either a single episode per forward or a batch of 20 episodes in parallel. We run the method in evaluation mode via `torch.no_grad()`. Timing is measured by injecting cuda events around function calls and measuring the elapsed time between them. The events are inserted via `torch.cuda.Event(enable_timing=True)`. We run our approach on a single NVIDIA V100 for 1000 episodes and report the average timings of each part.

# E  ADDITIONAL IMPLEMENTATION DETAILS

We provide code for the Gaussian Process inference, the neural network layers that make up the modules used in our approach, and the details of how the predictive output distribution from the GP is fed to the decoder.

## E.1  CODE FOR GAUSSIAN PROCESS

Pseudo-code for the dense GP is shown in Listing 1. Equation 7 involves the multiplication with a matrix inverse. It is in practice computed via the Cholesky decomposition and solving the resulting systems of linear equations. For brevity and clarity, we omit device casting and simplify the solve implementation. In practice, we use the standard triangular solver in PyTorch, `torch.triangular_solve`.

```
def GP(x_q, y_s, x_s, sigma_y, kernel):
    """ Produces the predictive posterior distribution of the GP.
    After each line, we comment the shape of the output, with sizes
    defined as:
    B is the batch-size
    Q is the number of query feature vectors in the query image
    S is the number of support feature vectors across the support images
    M is the number of channels in the GP output space
    D is the number of channels in the feature vectors.

    Args:
    x_q: deep query features (B,Q,D)
    y_s: support outputs (B,S,M)
    x_s: deep support features (B,S,D)
    sigma_y: mask standard deviation
    kernel: the kernel function,
    """
    B, S, D = x_s.shape
    I = torch.eye(S)
    K_ss = kernel(x_s, x_s)   #(B,S,S)
    K_qq = kernel(x_q, x_q)   #(B,Q,Q)
    K_sq = kernel(x_s, x_q)   #(B,Q,S)
    L_ss = torch.cholesky(K_ss + sigma_y**2 * I)   #(B,S,S)
```

Table 17: All neural network blocks used by our approach. The rightmost column shows the dimensions of the output of each block, assuming a $512 \times 512$ input resolution. The image encoder is from He et al. (2016) and the decoder from Yu et al. (2018). The `BottleNeck` and `BasicBlock` blocks are from He et al. (2016), and the `CAB` and `RRB` blocks from Yu et al. (2018). See their works for additional details.

| | Image Encoder | |
|---|---|---|
| conv1 | Conv2d | $64 \times 256 \times 256$ |
| bn1 | BatchNorm2d | $64 \times 256 \times 256$ |
| relu1 | ReLU | $64 \times 256 \times 256$ |
| maxpool | MaxPool2d | $64 \times 128 \times 128$ |
| layer1 | 3x BottleNeck | $256 \times 128 \times 128$ |
| layer2 | 4x BottleNeck | $512 \times 64 \times 64$ |
| layer3 | 6x/23x BottleNeck | $1024 \times 32 \times 32$ |
| layer4 | 3x BottleNeck | $2048 \times 16 \times 16$ |
| layer3 out | Conv2d | $512 \times 32 \times 32$ |
| layer4 out | Conv2d | $512 \times 32 \times 32$ |
| | **Mask Encoder** | |
| conv1 | Conv2d | $16 \times 256 \times 256$ |
| bn1 | BatchNorm2d | $16 \times 256 \times 256$ |
| relu1 | ReLU | $16 \times 256 \times 256$ |
| maxpool | MaxPool2d | $16 \times 128 \times 128$ |
| layer1 | BasicBlock | $32 \times 64 \times 64$ |
| layer2 | BasicBlock | $64 \times 32 \times 32$ |
| layer3 | BasicBlock | $64 \times 16 \times 16$ |
| layer2 out | Conv2d+BatchNorm2d | $64 \times 32 \times 32$ |
| layer3 out | Conv2d+BatchNorm2d | $64 \times 16 \times 16$ |
| | **Decoder** | |
| rrb in 1 | RRB | $256 \times 16 \times 16$ |
| cab 1 | CAB | $256 \times 16 \times 16$ |
| rrb up 1 | RRB | $256 \times 16 \times 16$ |
| upsample1 | Upsample | $256 \times 32 \times 32$ |
| rrb in 2 | RRB | $256 \times 32 \times 32$ |
| cab 2 | CAB | $256 \times 32 \times 32$ |
| rrb up 2 | RRB | $256 \times 32 \times 32$ |
| upsample2 | Upsample | $256 \times 64 \times 64$ |
| rrb in 3 | RRB | $256 \times 64 \times 64$ |
| cab 3 | CAB | $256 \times 64 \times 64$ |
| rrb up 3 | RRB | $256 \times 64 \times 64$ |
| upsample3 | Upsample | $256 \times 128 \times 128$ |
| rrb in 4 | RRB | $256 \times 128 \times 128$ |
| cab 4 | CAB | $256 \times 128 \times 128$ |
| rrb up 4 | RRB | $256 \times 128 \times 128$ |
| conv out | Conv2d | $2 \times 128 \times 128$ |
| upsample4 | Upsample | $2 \times 512 \times 512$ |

```
24    mu_q = (K_sq.T @ solve(L_ss.T, solve(L_ss, y_s))  #(B,Q,M)
25    v = solve(L_ss, K_sq)  #(B,S,Q)
26    cov_q = K_qq - v.T @ v  #(B,Q,Q)
27    return mu_q, cov_q
```
Listing 1: PyTorch implementation of the Gaussian Process utilized in the proposed approach. Here, the learning and inference is combined in a single step. The `@` operator denotes matrix multiplication and `solve` the solving of a linear system of equations. The `.T` is the batched matrix transpose.

### E.2 TRAINABLE LAYER LIST

In Table 17 we report the trainable neural network layers used in our approach. The image encoder listed is either a ResNet50 (He et al., 2016) or a ResNet101 (He et al., 2016) with two projection layers. The results of *layer3* and *layer4* are each fed through a linear projection layer, *layer3 out* and *layer4 out* respectively, to produce feature maps at stride 16 and stride 32. These two feature maps are then fed into one GP each in the GP pyramid. The support masks is fed through another ResNet (He et al., 2016) in order to produce the support outputs. The GPs are integrated as neural network layers, but do not contain any learnable parameters. The decoder is a DFN (Yu et al., 2018). We do not adopt the border network used in their work and we skip the global average pooling. We also feed it shallow feature maps extracted from the query. The shallow feature maps are the results of *layer1* and *layer2* in the image encoder, at stride 4 and 8 respectively.

### E.3 Final Mask Prediction Details

In 3.5 we transformed the output of the GP before presenting it to the decoder, restoring spatial structure. We present the equations for doing so. Let $(\cdot)$. denote tensor indexing. The mean representation fed to the decoder is found as

$$(\mathbf{z}_\mu)_{h,w} = (\boldsymbol{\mu}_{\mathcal{Q}|\mathcal{S}})_{hW+w}, \quad \mathbf{z}_\mu \in \mathbb{R}^{H \times W \times E} . \tag{10}$$

For the covariance we consider spatial neighbours within a square $N \times N$ window. Let $\delta = (N-1)/2$ and $(i,j) \in \{-\delta, \dots, \delta\}^2$ be the possible offsets within the window. Then, the covariance representation fed to the decoder is

$$(\mathbf{z}_\Sigma)_{h,w,(i+\delta)(2\delta+1)+(j+\delta)} = (\boldsymbol{\Sigma}_{\mathcal{Q}|\mathcal{S}})_{hW+w,(h+i)W+(w+j)}, \quad \mathbf{z}_\Sigma \in \mathbb{R}^{H \times W \times (2\delta+1)^2} . \tag{11}$$

## F Qualitative Results

We provide qualitative results on PASCAL-$5^i$ and COCO-$20^i$. First, we compare the our final approach to a baseline on the COCO-$20^i$ benchmark. The baseline also relies on dense GPs - but uses a linear kernel, does not utilize the predictive covariance or learn the GP output space, and uses a single feature level (stride 32). Our final approach instead adopts the SE kernel, adds the predictive covariance in a local $5 \times 5$ region, learns the output space, and employs dense GPs at two feature levels (stride 16 and stride 32). The results are shown in Figure 4. Our final approach significantly outperforms the baseline in these examples, making only minor mistakes.

In Figure 5 we show qualitative results of our final approach on the PASCAL-$5^i$ dataset. Our approach accurately segments the class of interested, even details such as sheep legs. In Figure 6, we show results on COCO-$20^i$.

We provide a qualitative comparison between different support set sizes in Figure 7. For this comparison, we use COCO-20. In general, the 1-shot setting is quite challenging for several classes due to major variations in where the object appears and what it looks like. If the single support sample is too different from the query sample, the model tends to struggle. With five support samples, the model performs far better. At five, the benefit of additional support samples seems to saturate, with ten samples often bringing only marginal improvements. An example is the train episode in Figure 7 where minor improvements are obtained when going from five to ten support samples.

| Support | Query | Baseline | Ours |
|---|---|---|---|

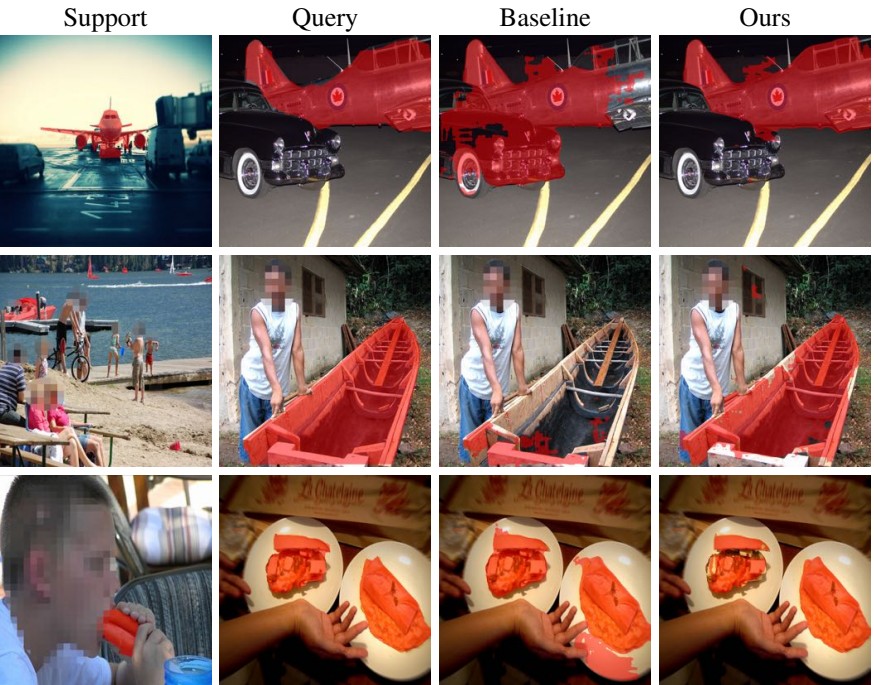

Figure 4: Qualitative comparison between our final model and baseline in the 1-shot setting from the COCO-20$^i$ benchmark. Human faces have been pixelized in the visualization, but the model makes predictions on the non-pixelized images.

Support Set          Query          Ours

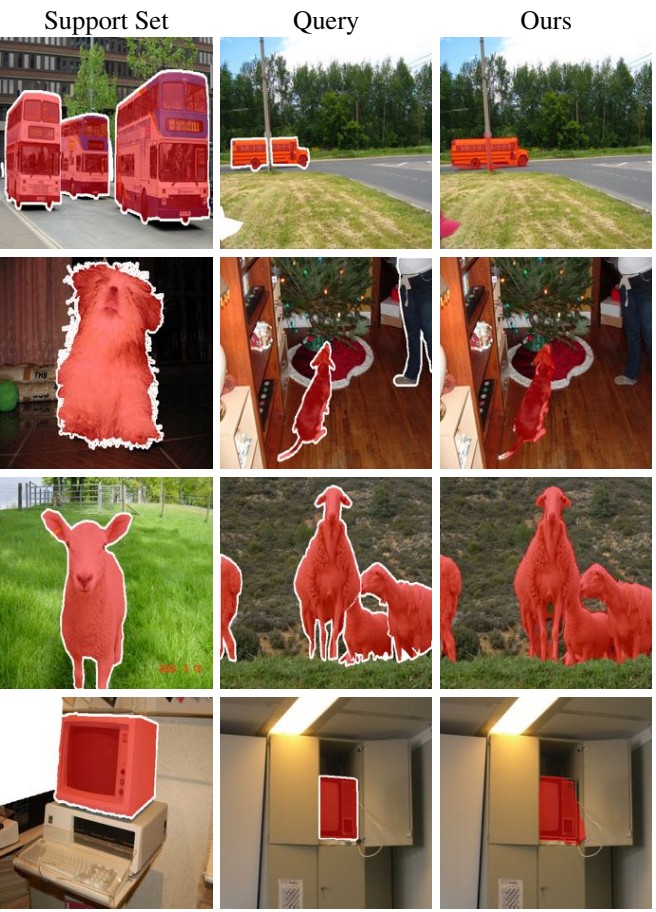

Figure 5: Qualitative results on challenging episodes in the 1-shot setting from the PASCAL-$5^i$ benchmark.

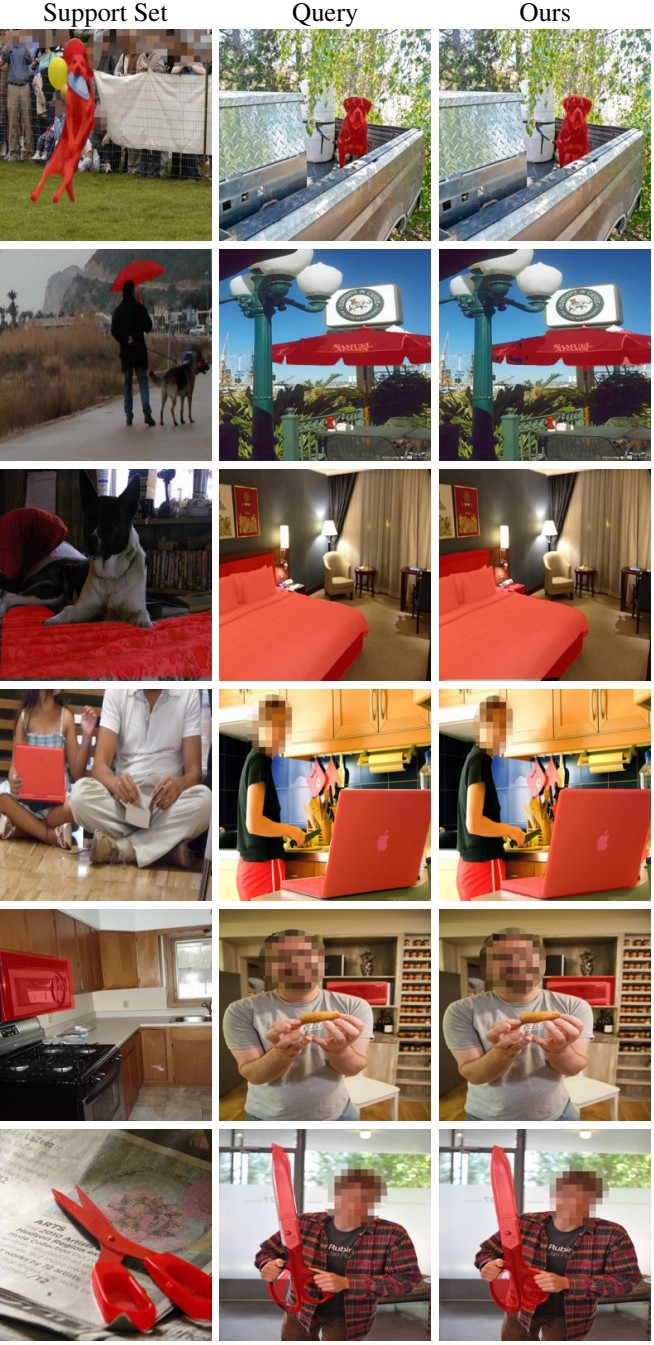

Figure 6: Qualitative results on challenging episodes in the 1-shot setting from the COCO-20$^i$ benchmark. Human faces have been pixelized in the visualization, but the model makes predictions on the non-pixelized images.

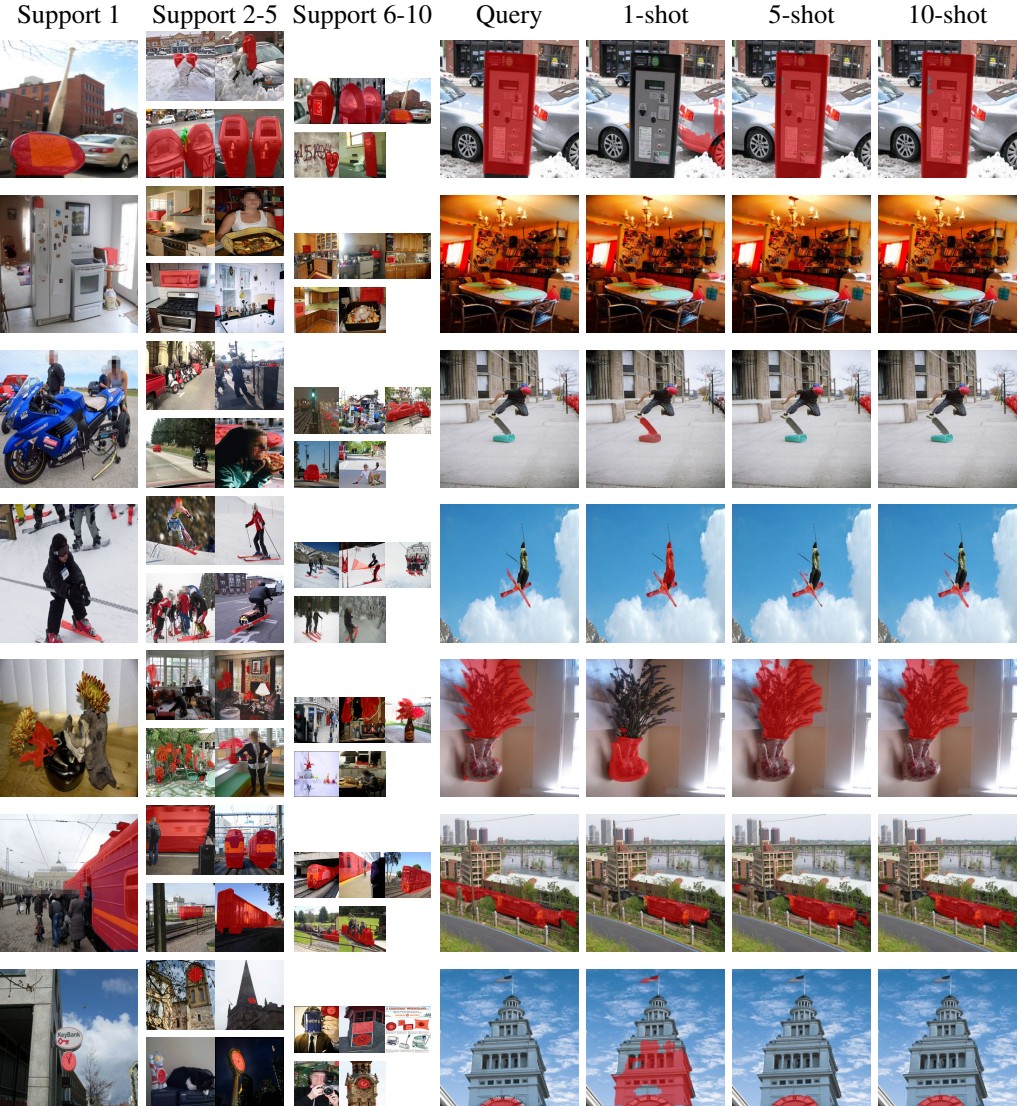

Figure 7: Qualitative results of our approach given 1, 5, and 10 support samples. The 1-shot results are based on Support 1; the 5-shot results on Support 1 and Support 2-5; and the 10-shot Support 1, Support 2-5, and Support 6-10. The results are from COCO-$20^i$ and human faces have been pixelized in the visualization, but the model makes predictions on the non-pixelized images.

