# OpenReview forum: "Dense Gaussian Processes for Few-Shot Segmentation"
_ICLR.cc/2022/Conference — ICLR 2022 Submitted_

### Official Review · Reviewer_Kjwm · 2021-11-02

**Correctness:** 3
**Technical Novelty And Significance:** 3
**Empirical Novelty And Significance:** 3
**Recommendation:** 6
**Confidence:** 5

**Main Review:**

Pro:
- A novel idea to combine GP and FSS.
- Extensive experiments to validate the results on different benchmarks
- State-of-the-art on two benchmarks, and gracefully extend to high-shot segmentation.


Con:
- Hyper-parameters sensitivity. Are the methods sensitive to different hyper-parameter in GP? The author should mention this in the paper.
- What is your implementation without Cov and GPO in Table 4? The detail is missing here.
- For me, it is still not clear how the input are interplayed in encoder.(E.3 only introduced about restoring spatial structure of the output of the GP)
- In contribution part, mentioned the probabilistic modeling, I don't see its importance in experimental parts.
- Visualization for 5 shot is not provided, what's the main improvement from 1shot to 5shot to 10shot?
- How about removing the contribution of f branch, only keeping shallow x?

**Summary Of The Paper:**

The paper strives for solving the few-shot segmentation problem, they propose to incorporate the Gaussian Process(GP) into the framework of few-shot segmentation. Except that, they also exploit the high-dimensional output space for GP, the result reaches state-of-the-art in two benchmarks,  one bonus is that  the segmentation quality scales gracefully as increasing the support set size.

**Summary Of The Review:**

The paper overall presents a novel idea with solid empirical validations. Except some minor concern I raised in previous parts. In total, I am inclined to accept this paper.  I am happy to raise the score if my concern is fully resolved.

---

> ### Author Response · Authors · 2021-11-21
> **Response to Kjwm**
>
> We thank this reviewer for the positive, thorough, and constructive feedback.
>
> ### **W1:** GP Hyperparameters
> We thank the reviewer for this suggestion. In table 3, we experimented with different kernels. Here, we supply experiments on PASCAL-$5^i$ where $\sigma_y^2$, $\sigma_f^2$, and $\ell^2$ are varied. To cover an order of magnitude, we multiply the hyperparameters with $0.3$ and $3.0$. The results are shown below. DGPNet is not sensitive to hyperparameters. One possible explanation is that the input to the GP and the interpretation of the GP output are both trained. For instance, changes to $\ell^2$ can be counteracted by scaling the output of the image encoder. We have added this experiment to the revised version of the appendix C.
>
> | $\ell^2 = 0.3\sqrt{D}$ | $\ell^2 = \sqrt{D}$ | $\ell^2 = 3.0\sqrt{D}$ |
> | -- | -- | -- |
> | 72.9 | **73.5** |  72.8 |
>
> | $\sigma_f^2 = 0.3$ | $\sigma_f^2 = 1.0$ | $\sigma_f^2 = 3.0$ |
> | -- | -- | -- |
> | 72.8 | **73.5** | 73.6 |
>
> | $\sigma_y^2 = 0.03$ | $\sigma_y^2 = 0.1$ | $\sigma_y^2 = 0.3$ |
> | -- | -- | -- |
> | 73.0 | **73.5** | 72.9 |
>
> ### **W2:** No Cov and GPO missing detail
> No Cov and No GPO is the same setup as in table 3, row "SE". Then we do not feed the covariance $z_\Sigma$ to the decoder and we do not learn the GP output space. Not learning the GP output space means that the mask-encoder $G$ performs average-pooling on the input mask to obtain $y_S$. No Cov and GPO means that we let the mask-encoder be a lightweight resnet that predicts 64-dimensional outputs $y_S$, but we do not use $z_\Sigma$.
>
> ### **W3:** How does input interplay?
> The input images $\\{I_{\mathcal{S}k}\\}\_k, I_{\mathcal{Q}}$ are processed independently by the CNN $F$. Thus, the individual images do not interplay in the image encoder. The masks $\\{M_{\mathcal{S}k}\\}\_k$ are processed independently by the mask encoder $G$. They do not interplay with each other or any of the images. These will be fed into the DGP-module in which they will interplay. The input is constructed as $(x_{\mathcal{S}})\_{kHW+hW+w} = (F(I_{\mathcal{S}k}))\_{h,w}$, $(x_{\mathcal{Q}})\_{hW+w} = (F(I_{\mathcal{Q}}))\_{h,w}$, and $(y_{\mathcal{S}})\_{kHW+hW+w} = (G(I_{\mathcal{S}k}))\_{h,w}$. That is, the input to the DGP-module is obtained via flattening and stacking. We will add further clarifications in the final version.
>
> ### **W4:** Probabilistic Modelling
> Please note that the GP inherently is a probabilistic model. As one of its advantages, it provides a principled uncertainty estimate. This is analyzed in table 4, where the covariance is shown to improve the performance on PASCAL and COCO 1-shot and 5-shot.
>
> ### **W5:** 5-shot Visualization
> We appreciate the suggestion. The manuscript has been updated with a qualitative comparison between 1, 5, and 10 shots (figure 7, Appendix F).
>
> ### **W6:** Removing f-branch
> We thank the reviewer for this suggestion. This question was also posed by another reviewer. Here, we restate our answer. To further analyze the effectiveness of our GP module, we evaluate three variants: **(i)** We remove the GP (the f-branch) and let the decoder rely only on the shallow features. **(ii)** We replace the GP with the Prior-mask learning mechanism proposed in PFENet. **(iii)** We replace the GP with a prototype-based approach where the support features are mask-pooled and the result compared to the query features using cosine-similarity, similar to e.g. PANet. Results on PASCAL and COCO are reported in the table below. Our approach outperforms all baselines by a significant margin. This further validates the impact of our DGP module. We have added these results to appendix B.3 of the revised version.
>
> | Method  | PASCAL 1-shot | PASCAL 5-shot | COCO 1-shot | COCO 5-shot |
> | ------------- | ------------- | ------------- | ------------- | ------------- |
> | **(i)** No GP | 42.9 | 43.0 | 23.5 | 23.5 |
> | **(ii)** PFENet Prior mask | 54.0 | 54.7 | 35.8 | 37.2 |
> | **(iii)** Prototype | 57.5 | 63.0 | 41.5 | 49.9 |
> | **(Ours)** With GP | **63.5** | **73.5** | **45.0** | **56.2** |

---

> ### Author Response · Authors · 2021-12-10
> **Answers to all your valuable comments (e.g., hyperparameter sensitivity). We appreciate if you would check them and reconsider the decision.**
>
> Dear Reviewer Kjwm,
>
> Thanks for your valuable feedback! We address all of your concerns and a summary of our detailed response is provided below. We humbly request you to check it and reconsider the decision.
> - A hyperparameter analysis shows that the proposed approach is not overly sensitive
> - We elucidate on how the inputs interplay
> - Qualitative 1-shot, 5-shot, and 10-shot results have been provided side-by-side in the Appendix
> - An ablation study on the f-branch is provided

---

### Official Review · Reviewer_uRb5 · 2021-11-03

**Correctness:** 3
**Technical Novelty And Significance:** 3
**Empirical Novelty And Significance:** 2
**Recommendation:** 6
**Confidence:** 4

**Main Review:**

Strengths:
- The idea of adopting dense Gaussian Process (GP) regression to learn the mapping from local deep image features to mask values, as well as considering uncertainty for the final segmentation is interesting.
- The paper is well written and the methodology technically sound.
- Authors report extensive experiments and the results achieved are very competitive.

Weaknesses:
- The idea that utilizing Gaussian processes in the context of few-shot classification has been exploited in [r1,r2]. Although the proposed method focuses on the segmentation task, it seems like a dense classification setting of r1 and r2. Authors need to clarify the main differences.
- Does this method only focus on 1-way (binary segmentation) setting? Can it solve the multiple novel classes in a single episode?

**Summary Of The Paper:**

Authors propose a novel few-shot segmentation method by adopting dense Gaussian process (GP) regression to capture complex appearance distributions. To boot the performance, authors consider the uncertainty in the final segmentation. Authors exploit the end-to-end learning capabilities of the proposed method to learn a high-dimensional output space for the GP.
Authors report state-of-the-art results in two public few shot segmentation benchmarks.

**Summary Of The Review:**

The idea of adopting dense Gaussian Process (GP) regression to learn the mapping from local deep image features to mask values, as well as considering uncertainty for the final segmentation is interesting. Although the proposed method seems like a dense setting of the existing methods, the paper is well written and the results achieved are very competitive. I'd like to enhance my rating if the authors address all my concerns.

---

> ### Author Response · Authors · 2021-11-21
> **Response to uRb5**
>
> We thank this reviewer for the positive and valuable feedback.
>
> ### **W1:** Comparison to [r1,r2]
> We assume that with [r1,r2] the Reviewer refers to the works
>
> [Massimiliano Patacchiola, Jack Turner, Elliot J. Crowley, and Amos Storkey. Bayesian meta- learning for the few-shot setting via deep kernels. In Advances in Neural Information Processing Systems, 2020.]
>
> [Jake Snell and Richard Zemel. Bayesian few-shot classification with one-vs-each po ́lya-gamma augmented gaussian processes. In International Conference on Learning Representations, 2021.]
>
> Compared to these two works for few-shot classification, we have the following contributions:
>
> * We design a GP module for a dense prediction task, which brings several novel challenges.
> * Instead of having the GP as the final output, we employ it in the core of an encoder-decoder architecture. This is crucial for tractability, allowing the GP to operate on low-resolution features, while CNN decoder predicts high-quality masks based on the GP outputs (Section 3.1 and 3.5).
> * We propose to learn the output space of the GP using a mask encoder module (Section 3.4), leading to substantial performance improvements.
> * We motivate and integrate uncertainty cues for FSS by employing the GP neighborhood covariance as an input to the segmentation decoder (Section 3.2 and 3.5).
> * We design a pyramid architecture by applying our GP module at different feature levels (Section 3.6).
>
> ### **W2:** Multiple classes
> We follow the standard benchmarks in FSS, which only consider the binary segmentation setting. A straightforward extension of our DGPNet to the K-way setting is possible using the strategy, often employed in Video Object Segmentation [Pont-Tuset, Jordi, et al.]. In our case, this would only lead to several of forward passes through the GP module and the decoder. Using the detailed timings in table 14, these steps constitute $27\%$ of the inference time (batch size 1, 5-shot). Thus, a 5-way, 5-shot setting takes about $144$ ms for the full few-shot learning and inference. We consider further investigation of the multi-class scenario an interesting future research direction.
>
> References:
> Pont-Tuset, Jordi, et al. "The 2017 davis challenge on video object segmentation." arXiv preprint arXiv:1704.00675 (2017).

---

> ### Author Response · Authors · 2021-12-10
> **Answers to all your valuable comments (e.g., comparison to [r1,r2]). We appreciate if you would check them and reconsider the decision.**
>
> Dear Reviewer uRb5,
>
> Thanks for your valuable feedback! We address all of your concerns and a summary of our detailed response is provided below. We humbly request you to check it and reconsider the decision.
> - A comparison to [r1,r2] (the references were missing from the review, but we are fairly confident on which two works the reviewer referred to) has been provided.
> - We elaborate on multi-way segmentation

---

### Official Review · Reviewer_b84d · 2021-11-03

**Correctness:** 3
**Technical Novelty And Significance:** 2
**Empirical Novelty And Significance:** 2
**Recommendation:** 6
**Confidence:** 4

**Main Review:**

This paper proposes a dense Gaussian Process approach for few shot segmentation scenario. Overall promising results are achieved comparing to the recent efforts. On the other hand, the paper has some issues to be discussed below.
*lack of clarification of the major contributions. It seems the main contribution lies in being first to apply the combination of GPs and Neural Networks to the few short semantic segmentation task. It is yet unclear what is the main technical contribution in this paper, despite reading Sec.3 multiple times.
*It is also surprising that despite being a much simpler method as shown in Fig.2 when comparing to existing methods such as SAGNN (CVPR'21), and despite the internal Gaussian representation is very coarse-scale and lacking-of-details as in Fig.2, the final results of this paper are much better. Is there any intuition of why this is the case?
*The authors should also present more visual results of e.g. their 5-short segmentation in the appendix/supplementary. It also lacks visual evidence of the claimed uncertainty reasoning benefits.
*What is the computational cost/time complexity for the training/inference processes, respectively?
*The authors should provide the implementation publicly available. Otherwise it is difficult for others to validate and reproduce the same results and performance.


**Summary Of The Paper:**

This paper proposes a dense Gaussian Process approach for few shot segmentation scenario. Overall promising results are achieved comparing to the recent efforts.

**Summary Of The Review:**

many aspects are unclear

---

> ### Author Response · Authors · 2021-11-21
> **Response to b84d**
>
> We thank this reviewer for the constructive feedback.
>
> ### **W1:** Clarification of contributions
> We introduce several technical contributions in order to design a highly effective GP-based FSS approach:
>
> * Instead of having the GP as the final output, we employ it in the core of an encoder-decoder architecture. This is crucial for tractability, allowing the GP to operate on low-resolution features, while CNN decoder predicts high-quality masks based on the GP outputs (Section 3.1 and 3.5).
> * We propose to learn the output space of the GP using a mask encoder module (Section 3.4), leading to substantial performance improvements.
> * We motivate and integrate uncertainty cues for FSS by employing the GP neighborhood covariance as an input to the segmentation decoder (Section 3.2 and 3.5).
> * We introduce a pyramidal architecture design by applying our GP module at different feature levels (Section 3.6).
>
> The effectiveness of our contributions are thoroughly analyzed in our experiments. Moreover, our final approach achieves a remarkable $+30\%$ relative improvement upon previous state-of-the-art on COCO 5-shot.
>
>
> ### **W2:** Intuition for why our approach is better
> We think there are a few strong reasons that explain the superior results of our approach.
>
> * Compared to standard neural network layers, our GP-module is capable of performing more complex reasoning. The final GP equations (6)-(7) used to implement the layer itself, are the result of a very rich theory for probabilistic modeling (see e.g. Rasmussen or Murphy for details). It can therefore integrate information from the support set in a very powerful manner.
> * Since the GP-module mainly relies of feature comparisons and directly outputs a mask encoding (instead of appearance features), we believe that our approach is less susceptible of meta-overfitting to the categories in the training set.
> * Our approach explicitly integrates uncertainty cues, which we argue are important for FSS (see Section 3.2).
> * While the GP operates on a coarse scale, it predicts a high-dimensional learned mask encoding (Section 3.4), which can capture details in the segmentation mask. This is then processed by the deep segmentation decoder (Equation 9), which further integrates shallow features to achieve an accurate mask prediction. Note that in Figure 2, we use a 1-dimensional mask encoding only for visualization purposes, while our final approach employs the multi-dimensional learned encoding (Section 3.4).
>
> References:
>
> Gaussian Processes for Machine Learning. Carl Edward Rasmussen and Christopher K. I. Williams. The MIT Press, 2006. ISBN 0-262-18253-X
>
> Murphy, Kevin P. Machine learning: a probabilistic perspective. MIT press, 2012.
>
> ### **W3:** Qualitative results for 5-shot
> We appreciate the suggestion. The manuscript has been updated with a qualitative comparison between 1, 5, and 10 shots (Figure 7, Appendix F).
>
> ### **W4:** Computational cost
> The computational cost for the inference is provided in appendix D and table 14. The training takes the longest for the COCO 5-shot experiments, around 10 hours on an A100. At 40 epochs à 1000 iterations, this corresponds to about a second per iteration.
>
> ### **W5:** Public implementation
> We have added our code to the supplementary material in order to make it public and will also release it on GitHub.

---

> ### Author Response · Authors · 2021-12-10
> **Answers to all your valuable comments (e.g., intuition). We appreciate if you would check them and reconsider the decision.**
>
> Dear Reviewer b84d,
>
> Thanks for your valuable feedback! We address all of your concerns and a summary of our detailed response is provided below. We humbly request you to check it and reconsider the decision.
> - We clarify the technical contributions
> - We provide our intuition on the high performance resulting from the GP-module
> - Qualitative 1-shot, 5-shot, and 10-shot results have been provided side-by-side in the Appendix.
> - The implementation has been made publicly available

---

### Official Review · Reviewer_nhTR · 2021-11-03

**Correctness:** 3
**Technical Novelty And Significance:** 2
**Empirical Novelty And Significance:** 2
**Recommendation:** 5
**Confidence:** 5

**Main Review:**

- Strengths:

    1. It's a solid idea to introduce uncertainty modeling into few-shot segmentation.

    2. The exact implementation details of the experiments are provided. Although the source code is not presented, the authors show its pseudo-code in SM.



- Weaknesses:

    1. The way of the GP outputs being used is somehow confusing. The authors claim that \mu_{Q|S} and \Delta_{Q|S} are the predicted mean and covariance of the mask of query, i.e., y_Q. However, the prediction of y_Q is fulfilled through DFN (a deep CNN), which means it's not ensured that the predicted \hat{y}_Q can be sampled from a Gaussian distribution with a relatively high probability.

    2. Since the input of the decoder is a concatenation of the GP outputs and shallow-level query image features, it's necessary to conduct another ablation study to show the influence on the predicted result cast by the GP outputs, i.e., the effectiveness of introducing the GP outputs.

   3. The use of 1-4 background-foreground weighting in the loss function actually performs hard-example mining, which is empirically found beneficial in binary segmentation tasks. As the baseline has already surpassed many of the other methods, the author should apply the proposed method on a standard training scheme, such as the SGD optimizer with the CE loss without weighting, following the practice of PANet, PPNet, PFENet and CANet.

   4. The inference speed should be discussed since an additional network DFN is appended as the decoder.

**Summary Of The Paper:**

This paper proposes a special Gaussian process (GP) named dense GP, to model a mapping between dense local deep features and their corresponding mask values. Based on this dense GP, a few-shot segmentation method named DGPNet is proposed.
The authors claim that DGPNet is novel in that it can be applied to situations that unseen classes are not linearly separable, and can produce the uncertainty of its prediction as well. To support this they conduct series of experiments on PASCAL-5^i and COCO-20^i.

**Summary Of The Review:**

This paper is novel mostly in that it utilizes a Gaussian process to generate an extra coarse prediction of the query image. However, an ablation study is lacking to prove the effectiveness of the dense GP module alone.

---

> ### Author Response · Authors · 2021-11-21
> **Response to nhTR**
>
> We thank this reviewer for the thorough and valuable feedback.
>
> ### **W1:** Clarification of GP outputs
> $\mu_{Q|S}$ and $\Sigma_{Q|S}$ is the mean and covariance of the output, $y_Q$. In our approach, the GP output $y_Q$ is the mask encoding (Section 3.4), which is not the same as the predicted mask of the query, $M_Q$. Instead, the final mask output $M_Q$ is generated by the DFN decoder $M_Q = U(z_\mu, z_\Sigma, x_{\mathcal{Q}, \text{shallow}})$, taking the GP $\mu_{Q|S}$ and $\Sigma_{Q|S}$ (reshaped as $z_\mu, z_\Sigma$, see Section E.3) along with shallow features $x_{\mathcal{Q},\text{shallow}}$. Thus, we do not intend the final mask output $M_Q$ to be Gaussian. As one of our contributions, the GP output space $y_Q$ is learned by the encoder network in a data-driven manner (Section 3.4). We will further clarify these aspects in the revised version.
>
> ### **W2:** Ablation of GP-module
> We thank the reviewer for this suggestion. To further analyze the effectiveness of our GP module, we evaluate three variants: **(i)** We remove the GP (the f-branch) and let the decoder rely only on the shallow features. **(ii)** We replace the GP with the Prior-mask learning mechanism proposed in PFENet. **(iii)** We replace the GP with a prototype-based approach where the support features are mask-pooled and the result compared to the query features using cosine-similarity, similar to e.g. PANet. Results on PASCAL and COCO are reported in the table below. Our approach outperforms all baselines by a significant margin. This further validates the impact of our DGP module. We have added these results to appendix B.3 of the revised version.
>
> | Method  | PASCAL 1-shot | PASCAL 5-shot | COCO 1-shot | COCO 5-shot |
> | ------------- | ------------- | ------------- | ------------- | ------------- |
> | **(i)** No GP | 42.9 | 43.0 | 23.5 | 23.5 |
> | **(ii)** PFENet Prior mask | 54.0 | 54.7 | 35.8 | 37.2 |
> | **(iii)** Prototype | 57.5 | 63.0 | 41.5 | 49.9 |
> | **(Ours)** With GP | **63.5** | **73.5** | **45.0** | **56.2** |
>
>
>
> ### **W3:** Impact of loss weighting and optimizer
> We selected AdamW over SGD since it is less sensitive to hyperparameters and is the de-facto standard for attention-based architectures (DETR, ViT, DeIT), which are reminiscent of our GP module (as mentioned in section 3.4 in the paper). We also considered loss weighting to be standard procedure, as it is performed in DENet and RePri to counter the class-imbalance. As suggested by the reviewer, we further ablate these choices in the table below. We compare to the methods leading either setting (see Table 1 in the paper). We report results using the ResNet-50 backbone, with the exception of SAGNN on COCO, where only ResNet-101 results are available (marked with $^*$). Replacing AdamW with SGD only leads to a marginal drop in performance. Furthermore, we found the loss weighting to primarily affect COCO 1-shot results. However, our approach still achieves the best result in this setting, i.e. SGD and no weighting on COCO 1-shot. When comparing to previous methods with loss weighting (DENet and RePri), our weighted approach with SGD achieves substantially better performance in all cases. In the 1-shot case, ours with SGD and no weighting only falls marginally behind (-0.4) SAGNN on PASCAL, while outperforming it by a large margin of +3.8 on COCO. Lastly, note that even our approach with SGD and no weighting outperforms all prior works in the 5-shot case by remarkable gains of over 6.0 on PASCAL and 11.1 on COCO.
>
>
> | Method  | Optim. | L-Weight | PASCAL 1-s | PASCAL 5-s | COCO 1-s | COCO 5-s |
> | -- | -- | -- | -- | -- | -- | -- |
> | CANet    | SGD   |    | 55.4 | 57.1 | 40.8 | 42.0 |
> | DENet    | SGD   | X  | 60.1 | 60.5 | 42.8 | 43.0 |
> | SAGNN    | SGD   |    | 62.1 | 62.8 | 37.2$^*$ | 42.7$^*$ |
> | RePri    | SGD   | X  | 59.1 | 66.8 | 34.0 | 42.1 |
> | -- | -- | -- | -- | -- | -- | -- |
> | **Ours** | SGD   |    | 61.7 | 72.8 | 41.0 | 54.1 |
> | **Ours** | SGD   | X  | 61.9 | 72.2 | 44.3 | 55.2 |
> | **Ours** | AdamW |    | 61.9 | 73.4 | 41.8 | 55.7 |
> | **Ours** | AdamW | X  | 63.5 | 73.5 | 45.0 | 56.2 |
>
>
> ### **W4:** Inference speed
> The inference speed is provided in appendix D and table 14. With a batch size of 1 and the 5-shot setting, the decoder is fairly lightweight, taking around 10\% of the computation time.

---

> ### Author Response · Authors · 2021-12-10
> **Answers to all your valuable comments (e.g., GP ablation). We appreciate if you would check them and reconsider the decision.**
>
> Dear Reviewer nhTR,
>
> Thanks for your valuable feedback! We address all of your concerns and a summary of our detailed response is provided below. We humbly request you to check it and reconsider the decision.
> - An ablation of the GP outputs shows the effectiveness of the GP-module.
> - An optimizer/loss-weighting ablation. Even without the foreground-background weighting and with a different optimizer, the proposed approach performs on-par with the state-of-the-art in 1-shot, and sets a new state-of-the-art in 5-shot.
> - The inference speed is provided in Appendix D.

---

### Author Response · Authors · 2021-11-21
**Thank you for the feedback**

We thank all reviewers: nhTR, b84d, uRb5, Kjwm, for their positive feedback. The reviewers found the method to be novel (Kjwm and nhTR) and interesting (uRb5). The reviewers also appreciated the extensive experiments (uRb5 and Kjwm), and acknowledge that the method achieves promising (b84d), very competitive (uRb5), and state-of-the-art (uRb5, Kjwm) results, with graceful scaling to additional support samples (Kjwm). Finally, the reviewers point out that the paper is well written and that the methodology technically sound (uRb5). We address all reviewers' questions and concerns below.

**Source code** We have added the source code as a zip-file in the supplementary material.

---

### Author Response · Authors · 2021-12-10
**Summary of our responses**

We have addressed the reviewers’ concerns and posted a summary under each reviewer. If there are any further questions, we are happy to answer them.

---

### Decision · Program_Chairs · 2022-01-20

**Decision:**

Reject

**Comment:**

This paper proposes the use of Gaussian process regression embedded into a neural network architecture for few-shot segmentation. In more detail, support and query images and support masks are fed through their encoders and their corresponding features are then used for Gaussian process regression to infer the distribution of the query mask encoding given the support set and the query images. The mean and the variance characterizing the GP predictive distribution is then fed into a CNN-based decoder to make the final prediction (segmentation).  The method is evaluated on PASCAL-5^i and COCO-20^i datasets, showing the superiority of the proposed approach wrt several competitive baselines.

Overall, the reviewers found the approach of using GPs within the proposed architecture interesting and somewhat significant and novel to the few-shot segmentation community. Technically, the proposed method does not develop a new algorithm and simply uses standard Gaussian process regression. The authors seemed to have addressed several concerns raised by the reviewers including the ablation study evaluating the influence of the GP module. However, the reviewers felt that there were quite a few changes/clarifications to the paper and new results that were not highlighted in the revised version, which made it difficult to provide a new assessment of the paper. Furthermore, the reviewers also thought that the authors did not provide convincing explanations in terms of the improvements from 1-shot to 5-shots, the not-so-good results when the model was trained with standard SGD without loss weighting and the rationale behind the success of the 5-shot setting.